# Prosocial decisions in naturalistic helping scenarios are predicted by cost-benefit tradeoffs and individual disposition
Qianying Wu [1,2] ✉, Miao Song [1], Jackie Ayoub[1], David Dunning [3] ✉, Danyang Tian[1] & Ehsan Moradi-Pari [1] ✉

The origins of human prosociality, in particular between strangers, are multifaceted. While laboratory studies support a cost-benefit account of helping, real-life scenarios involve additional socio-emotional motives grounded in subjective intuitions. How the cost-benefit model generalizes to everyday helping behavior remains unclear. In this study, we comprehensively assessed how motivations jointly shape helping across 100 naturalistic helping scenarios: an online sample ($N_1 = 215$) rated willingness to help after reading brief vignettes, and a subset ($N_2 = 140$) rated the strengths of candidate motivations elicited by each scenario. Two key factors—benefit to both helper and helpee, and cost to the helper—were identified through a factor analysis of the motivation ratings. We then successfully predicted helping decisions as a linear weighted sum of the two motivational factors, along with a dispositional helping bias. While a higher helping bias was associated with greater trait agreeableness and dispositional empathy, whereas individuals who prioritized cost over benefit exhibited higher levels of punishment sensitivity. Finally, we characterized the helping scenarios in three associated spaces: a decision space (willingness to help levels), a motivation space (two key motivational factors), and a semantic space (14 semantic types). Combining computational modeling with naturalistic helping contexts, this approach provides an integrated account of prosocial motivation and clarifies how individual differences in personality map onto real-world helping behaviors.

Prosocial behaviors, where individuals act to benefit others at a personal cost, occur frequently in daily life and are foundational to social cohesion and societal well-being[1]. Helping is a common form of prosocial behavior that involves goal-directed actions aimed at addressing others' needs[2]. Whereas evolutionary accounts have explained such behavior in terms of kin selection[3], human helping extends well beyond genetic or emotional ties to include aid toward strangers. Why humans help strangers and how such decisions are made has therefore received significant attention[4].

Traditional economic theories frame decision-making as a cost-benefit tradeoff, assuming that individuals prioritize self-interest by maximizing personal benefit and minimizing cost[5,6]. Later research extended this framework by showing that individuals also consider others' welfare in decisions: the subjective utility of helping has been successfully modeled as an integration of self- and other-related outcomes[7–14]. For instance, one study modeled decisions to forgo monetary rewards (cost) to prevent others from

hearing aversive noise (benefit)[11], while another examined effort exertion (cost) to earn money for others (benefit)[13]. Although these studies provided rigorous support for cost-benefit models, they relied on artificial, repetitive helping scenarios and objectively controlled costs and benefits. Such simplifications, albeit necessary for internal validity, limit our understanding of how cost-benefit tradeoffs operate in everyday settings.

To address this gap, we developed a task grounded in naturalistic helping scenarios. Our aim was two-fold: (1) to test whether cost-benefit computations generalize to real-world situations, and (2) to identify domain-specific motivational mechanisms that emerge in prosocial contexts. Our design enhances ecological validity[15] along two key axes: from artificiality to naturality, and from simplicity to complexity.

Moving from artificiality toward naturality, we emphasize that real-world helping often relies on rich contextual knowledge rather than explicit numerical information. Most laboratory paradigms intentionally minimize

[1]Honda Research Institute USA, San Jose, CA, USA. [2]California Institute of Technology, Pasadena, CA, USA. [3]University of Michigan Ann Arbor, Ann Arbor, MI, USA. ✉e-mail: qwu@caltech.edu; ddunning@umich.edu; emoradipari@honda-ri.com

such priors by using abstract, well-defined tradeoffs (e.g., exact amounts of money or effort)[15,16]. In contrast, daily helping requires drawing on common-sense understanding of human behavior and social situations[17]. To approximate this, we created a stimulus set of 100 brief narratives depicting diverse helping scenarios, primarily involving strangers. Instead of assigning fixed monetary or physical costs, we collected subjective ratings across a wide array of motivational dimensions to capture intuitive judgments in natural settings.

Moving from a simplicity-focused approach to embracing real-world complexity, we expanded the definition of costs and benefits beyond domain-general valuation currencies such as money and effort. As helping decisions largely involve social cognition and perception, we proposed that domain-specific motivations that are social and affective in nature play important roles in driving these decisions. These motivations reflect three broad theoretical traditions in the psychology of prosocial behavior. First, from an evolutionary perspective, helping unrelated individuals may promote long-term benefits through reciprocal altruism or competitive altruism, by fostering future cooperation or enhancing one's reputation and social status[4,18–21]. Second, emotional theories such as the 'negative-state relief model' emphasize internal affective benefits: helping alleviates personal distress caused by witnessing others suffer, mitigates guilt, and improves mood[22–24]. Third, socialization perspectives highlight the role of moral norms and social expectations, suggesting that people help others to maintain a positive self-concept, fulfill internalized social responsibilities, and avoid social punishments (e.g., self-concept threats, reputation damage)[25–28]. Although these motivations have each been examined in isolation, they are rarely considered together within a single computational framework.

Here, we hypothesized that helping decisions in everyday contexts still follow a cost-benefit tradeoff model but involve an expanded motivational architecture. We operationalized this using 12 motivational dimensions (guilt relief, self-concept maintenance, expected reciprocity, etc.), rated by a group of online participants. Costs and benefits associated with the scenarios were constructed through exploratory factor analysis and subsequently applied to a decision model to predict helping decisions collected from a broader sample.

To model these decisions, we built a computational framework adapted from a previous study[11]. The subjective utility of helping was modeled as a weighted combination of two components: a contextual component consisting of the cost and benefit of each helping situation (characterized by parameters $w_{cost}$ and $w_{benefit}$), and an additional fixed component describing an individual's dispositional helping propensity (characterized by parameter $b$). In accordance with previous research, we tested several model variants concerning different assumptions, including the use of dependent or independent weights for cost and benefit (i.e., whether emphasis on cost implies less emphasis on benefit), and the inclusion or exclusion of nonlinear effects (i.e., whether perceived cost or benefit is disproportionately scaled in the decision process).

Next, we leveraged this individualized modeling approach to examine how personality traits map onto distinct decision mechanisms. Although prior research has linked traits such as empathy and agreeableness to helping[29–31], these studies relied on a single outcome measure—an overall helping rate. Our framework allows for a fine-grained analysis of how specific personality factors modulate sensitivity to cost, benefit, and baseline helping propensity, offering a richer picture of individual differences in real-world helping behavior. More specifically, we hypothesized that a higher dispositional helping propensity (or bias of helping, $b$) is associated with higher agreeableness, higher dispositional empathy, and a more altruistic social value orientation; meanwhile, higher weights ($w_{cost}$) on cost and benefit ($w_{benefit}$) were expected to correlate with higher sensitivity to punishments and rewards, respectively (Table 1).

Finally, aggregating data over all participants, we characterized the diversity of these 100 helping scenarios among three latent spaces: a decision space reflecting group-level willingness to help (WTH) ratings, a motivational space representing perceived costs and benefits of helping, and a

semantic space clustering the scenarios into 14 categories (e.g., vulnerable group, crime, safety). Using a set of statistical analyses, we demonstrated how different contexts engage different motivations and ultimately relate to a broad spectrum of willingness to help. Our detailed characterization of everyday helping scenarios offers a valuable resource for future psychological research.

## Methods
### Participants
230 participants were recruited online through the Prolific crowd-sourcing facility (https://www.prolific.com) between August 2024 and September 2024. All participants completed Survey 1. After the quality check, we invited 150 of them to participate in Survey 2, and 144 of them completed Survey 2. Participants all passed the screening criteria through the screening tools from Prolific self-reports, including (1) United States as country of residence, (2) fluency in English, (3) age between 18 and 65 years, and (4) a 98% approval rate on past experiments. After exclusion based on data quality (see details in 'Data quality check and exclusion'), the final sample included 215 participants in Survey 1 (105 males, 110 females, age = 35.16 ± 10.86), and 140 participants in Survey 2 (68 males, 71 females, age = 34.42 ± 9.77). Detailed demographic characteristics of the two survey samples are shown in Table 2. The study is reviewed by the Institutional Review Board for Health Sciences and Behavioral Sciences (IRB HSBS) at the University of Michigan (protocol ID: HUM00260027), and deemed exempt from ongoing IRB review. All participants provided voluntary informed consent via an online consent form, and were compensated $10 for participating in each survey.

### Preregistration
The experiment plan and analysis plan were not preregistered prior to data collection.

### Everyday helping scenarios stimulus set
As short narratives have been proven effective in eliciting emotions and desires that are close to real-life experiences[32–34], here we created a text-based stimulus set consisting of 100 everyday helping scenarios. The stimuli are brief (33–76 words) descriptions of situations in which one or more people are in need of help from others (see a full list of scenarios in Supplementary Data 1). An example situation is as follows:

"You are doing your weekly grocery shopping in a large, sparsely populated store. You notice a person struggling to reach a can of tomato paste on the top shelf. They jump but fail to grab it. It's not high for you."

The stimuli were selected by aggregating three sources of scenarios: (1) a prior survey from the research team collecting ideas about circumstances when people think they are in need of help (Song et al., in preparation); (2) experimental stimuli from existing studies that asked participants to evaluate their willingness to help in real-life situations; and (3) brainstormed ideas from ChatGPT (GPT-4o). In all cases, a full scenario narrative was generated by expanding a core scenario idea (e.g., "help someone lift a heavy bag in the airport") to more detailed descriptions, including the helper's and helpee's ongoing activity, location of the event, features of the environment, etc. To avoid response biases induced by the helpee's demographic features (i.e., age and gender), we neutralized and omitted this information as much as possible (e.g., use of 'they/them' instead of 'he/him' and 'she/her', not revealing the age of the helpee unless they are minors).

### Everyday helping decision task
In the everyday helping decision task, each participant was presented with 50 different scenarios that were randomly selected from a total of 100 scenarios, and was asked to indicate their willingness to help (WTH) using a 6-point Likert scale (e.g., "How likely will you help the person grab the tomato paste?"—Not at all/ Very unlikely/ Somewhat unlikely/ Somewhat likely/ Very likely/ Absolutely likely). Participants were required to imagine

**Table 1 | Summary of hypotheses and expected statistical results**

|     | Hypothesis | Expected Results from Statistical Test |
| --- | --- | --- |
| H1a | Higher bias of helping is associated with higher agreeableness | Positive correlation between model parameter $b$ and the agreeableness subscore of BFI |
| H1b | Higher bias of helping is associated with higher dispositional empathy | Positive correlation between model parameter $b$ and empathetic concern subscore of IRI |
| H1c | Higher bias of helping is associated with more altruistic social value orientation | Positive correlation between model parameter $b$ and the SVO total score |
| H2a | Higher weight on cost is associated with higher punishment sensitivity | Positive correlation between model parameter $w_{cost}$ and BIS total score |
| H2b | Higher weight on benefit is associated with higher reward sensitivity | Positive correlation between model parameter $w_{benefit}$ and RR total score |

*Note that in the selected model, $w_{benefit} = 1 - w_{cost}$.

**Table 2 | Demographic and behavioral characteristics of survey samples**

|     | Survey 1 | Survey 2 | Comparison |
| --- | --- | --- | --- |
| N (before exclusion) | 215 (230) | 140 (144) | — |
| Sex (female:male) | 110:105 | 71:68 | $\chi^2(1) = 2.80 \times 10^{-6}, p = 0.999$ |
| Gender (woman:man:non-binary) | 106:104:3 | 68:68:1 | $\chi^2(2) = 0.003, p = 0.998$ |
| Age (M[SD]) | 35.16 [10.86] | 34.42 [9.77] | $t(353) = 0.650, p = 0.516$ |
| Race (white: Black: Asian: Native: multiracial) | 123:49:20:2:14 | 81:33:11:2:9 | $\chi^2(4) = 0.005, p = 0.999$ |
| Ethnicity (Hispanic/non-Hispanic) | 26:189 | 17:122 | $\chi^2(1) = 1.77 \times 10^{-5}, p = 0.997$ |
| Education (<high school: hs: some college: associate: bachelor: graduate) | 2:29:37:31:83:31 | 2:21:21:25:51:17 | $\chi^2(5) = 0.021, p = 0.999$ |
| Income (<25k: <50k: <75k: <100k: <150k: >150k) | 30:41:51:32:33:21 | 20:32:29:18:20:16 | $\chi^2(5) = 0.018, p = 0.999$ |
| Willingness to help (M[SD]) | 1.52 [1.62] | 1.53 [1.36] | $t(353) = -0.114, p = 0.909$ |
| % of help decisions (M[SD]) | 72.29% [19.84%] | 72.03% [17.57%] | $t(353) = 0.084, p = 0.933$ |

Summary of the participant demographic profiles (sex, gender, age, race, ethnicity, education level, income level) and task performance (willingness to help, % of help decisions) in Survey 1 and Survey 2, as well as statistical comparisons between the two surveys. All the variables are matched between two surveys. Participants who answered 'prefer not to say' or 'other' for some questions were not included in the comparisons of corresponding variables.

themselves in the scenario and witness the event. A specific helping option was clearly presented in the question. Importantly, to minimize confusion and avoid irrelevant psychological confounds, they were instructed to bear in mind the following assumptions:

- The person/people in need clearly expect to receive help from others.
- There is not a direct request from the person/people in need to the participant.
- The participant has enough capability/skills to help.
- If the participant's decision is not to help, nobody in the scenario will know their identity.

Given these assumptions, we expected participants to rate based on their voluntary decisions. We asked three quiz questions about these assumptions to guarantee their comprehension. Only if the participants answered all the quiz questions correctly were they allowed to move to the main task.

In addition, during each scenario, participants could indicate with a checkbox whether they had difficulty understanding or imagining the scenario to a degree that would affect their decision. Note that all scenarios were equally likely to be selected, thus the number of times that each scenario was rated was balanced.

**Helping motivation rating task**

To assess the contextual features of each scenario (such as costs and benefits) that potentially motivate one's helping or not helping decisions, we designed a comprehensive list including 15 scenario evaluation questions. The design and selection of the 15 questions were based on existing theories of helping motivations[20,21,23,26,28,35–39]. Supplementary Table 1 lists all the questions. Four questions (Q1-Q4) were evaluations of the characteristics of the scenario (e.g., empathy, urgency, etc.); seven questions (Q5-Q11) asked participants to imagine that they helped the people in need, and evaluate the cost (self

cost only) and benefit (both self and other's benefit) of helping; and the rest four questions (Q12-Q15) asked participants to imagine that they did not help the people in need (and the people in need did not receive any help) and evaluate the cost of not helping (i.e., bad consequences for the people in need, and bad feelings for self).

In the helping motivation rating task, each participant rated 20 different scenarios randomly selected (with equal probability) from all 100 scenarios. For each scenario, they first read the descriptions and rated on 15 questions sequentially. All the ratings were on 5-point Likert scales. They could also report additional motivations or considerations that affect their decisions in a text box. Similar to the decision task, participants had to pass 3 pre-task quiz questions to demonstrate their understanding of the instructions; and during the task, they could use a checkbox to indicate their inability to understand or imagine a specific scenario.

**Trait measures**

Using the helping decisions and ratings data, we sought to build individualized models that characterize one's helping behavior through a set of parameters. These parameters reflect dispositional individual differences that might correlate with previously established psychological trait dimensions. Therefore, we collected questionnaires to measure five potentially associated psychological traits. These included (1) Brief Interpersonal Reactivity Index (B-IRI) that has four subscales: empathic concern (EC), fantasy (FA), perspective taking (PT), and personal distress (PD), to measure dispositional empathy[40]; (2) Social Value Orientation (SVO) to measure how individuals weight the welfare of others compared to self[41]; (3) Reward Responsiveness (RR) to measure one's sensitivity to reward[42]; (4) Behavioral Inhibition System (BIS) scale to measure one's sensitivity to punishment[43]; and (5) 10-item abbreviated Big Five Inventory (BFI-10) that measures personality in 5

domains: extroversion, agreeableness, conscientiousness, neuroticism, and openness[44].

## Procedure

The entire study consisted of two survey sessions, and a subset of participants who completed Survey 1 were invited to participate in Survey 2.

In Survey 1, participants completed an everyday life helping decision task (hereafter referred to as 'decision task'). Participants first went through detailed instructions about the task, and passed several quiz questions to demonstrate their comprehension. Then they made decisions on 50 imagined helping scenarios. After the task, participants answered demographic questions (i.e., age, biological sex, gender, race, ethnicity, education level, and income level), and completed five questionnaires assessing various psychological traits. Survey 1 took 43 min on average.

In Survey 2, returning participants from Survey 1 performed the helping motivation rating task (hereafter referred to as 'rating task'). Similar to the decision task, participants first passed the instruction and comprehension quizzes, and then rated their feelings and experiences when imagining 20 different helping scenarios. Survey 2 took 55 min on average. Here, we invited participants from Survey 1 so that the decision and rating task data came from the same population (i.e., two samples were matched in demographics and helping decision responses, see details in Supplementary Table 3), thus we assumed the group-level consensus ratings applied to the Survey 1 sample.

Note that all participants completed Survey 1 before Survey 2. In this way, we guaranteed that the proposed motivational dimensions were not revealed to participants during their helping decisions, so that all the decisions were made based on participants' own considerations.

## Data quality check and exclusion

In the Survey 1 everyday helping decision task, we applied several quality control techniques. We implemented 3 attention check questions and 3 repeated scenario ratings across the scenarios. Participants who met any of the following criteria were excluded in further analyses and were not invited for Survey 2:

1. Failed to answer >=2 attention check questions
2. Gave >=2 inconsistent binary helping decisions (i.e., 6-point ratings were recoded to either help or not help, out of the 3 repeated scenarios)
3. The standard deviation of the ratings < 0.5 (i.e., an indication of straight line response)
4. Indicated 'don't understand or can't imagine the scenario' for >=10 scenarios.

Participants who passed the criteria ($N_1$ =215, 15 excluded) were re-invited to participate in Survey 2 (150 slots in total). In addition, at the scenario level (within each participant data), we excluded the scenarios in which participants indicated that they had difficulty in understanding or imagining (0.27% of responses).

In the Survey 2 helping motivation rating task, we first implemented 3 attention check questions. Participants who failed to answer >=2 attention check questions correctly were excluded for further analysis. Similar to Survey 1, if one indicated difficulty in understanding or imagining a scenario, all ratings for this scenario were excluded (0.39% of responses). Next, for each rater, we excluded their ratings for one question (across all 20 scenarios) if the ratings met either of the following criteria:

1. The standard deviation of the ratings across 20 scenarios < 0.1 (i.e., low response variability).
2. The mean rating exceeds 3 standard deviation above/below the gorup mean (i.e., outlier).

In total, we excluded 10 raters (6 did not complete the task, 4 lack of reliability) and 17 individual question ratings (from 7 participants). All the scenarios were rated by at least 25 independent raters (Mean = 27.69, s.d. = 1.23).

## Inter-rater reliability of the helping motivation ratings

We calculated the inter-rater reliability of each motivation feature rating using the intra-class correlation coefficient (ICC). We applied the two-way random ICC model to calculate the absolute agreement of the average rating score (ICC(A, k)) using the 'iccNA' function from the 'irrNA' package (version 0.2.3) in R (version 4.4.1)[45]. According to a common criterion, ICC <0.5 indicates poor reliability, 0.5~0.75 indicates moderate reliability, 0.75~0.9 indicates good reliability, and ICC >0.9 indicates excellent reliability[46].

## Exploratory factor analysis on the contextual motivation factors

During the rating task, we collected subjective evaluations of all scenarios across 15 motivational dimensions. We then derived consensus ratings for all the dimensions of all the scenarios by calculating the average rating across raters who passed the quality check. As such, we obtained the consensus ratings of all 15 motivational dimensions among 100 scenarios. These dimensions were intended to represent the contextual component in the helping decision model. However, since the questions were not designed to be independent or orthogonal to each other, some of them might characterize the same underlying construct, thus the dimensionality reduction was necessary.

To identify a reduced set of latent, parsimonious dimensions, we ran an exploratory factor analysis (EFA) on the ratings of 12 (out of the 15) motivation-related questions (i.e., empathy, descriptive norm, injunctive norm, self cost of help, other benefit of help, indirect reciprocity, mood management, reputation, positive self-concept, other cost of no help, guilt, negative self-concept). Note that Q11 and Q15 were summary questions of several proceeding questions (Q11 summarised Q7-Q10, Q15 summarised Q13-Q14), and Q2 (urgency) was considered as a separate environmental factor that was not directly related to inherent motivations, but rather modulates the willingness to help independently, therefore, they were not included in the EFA.

The number of factors was determined using a parallel analysis that identified factors whose eigenvalues are greater than those from a simulated distribution. As the parallel analysis suggested 2 factors, we conducted the EFA with a 2-factor structure using the varimax rotation (assuming an orthogonal factor structure). Note that the use of a promax rotation (that allows correlations between factors) generated highly consistent factor loadings (correlation between Varimax and Promax loadings across items: Factor 1: $R = 0.996$, Factor 2: $R = 0.984$) and also suggested minimal correlation ($R = 0.01$) between the 2 factors, thus in subsequent analyses, we proceeded with the varimax rotation results. Next, we extracted factor loadings on each item (i.e., motivation questions): an item belongs to a factor if its loading is the highest among other factors, and the absolute magnitude of the loading exceeds 0.3. Based on this criterion, we identified 2 factors: both benefit and self cost, that explained a total of 87% of the variance. The factor scores were estimated using the regression ("Thurstone") method in the "psych" package, and represented the magnitude of 'both benefits' and 'self cost' in the subsequent decision models, respectively. The EFA was performed using the 'fa' function from the "psych" package (version 2.4.6.26) in R (version 4.4.1).

## Associations between motivational dimensions and WTH

We assessed how well each motivation predicts the WTH across scenarios through linear mixed-effects models. As the motivational dimensions highly correlate with each other, to avoid the problem of multi-collinearity, we ran separate models for each motivation in the formula of $WTH = 1 + Motivation + (1 + Motivation|sub)$. The model included a motivation as a fixed effect and allowed for random intercepts and random slopes for each participant. We quantified the model fits using marginal $R^2$ (variance attributed

to fixed effects only) and conditional $R^2$ (variance attributed to both fixed effects and random effects)[47].

## Computational models of helping decisions

We first derived helping decisions after binarizing the WTH ratings into help (somewhat likely, very likely, absolutely) and not help (not at all, very unlikely, somewhat unlikely), because in the real life, any levels of WTH must translate to such a decision to affect actions. Then, to model each participant's decisions, we assumed that people compute a subjective utility of helping ($U$) in each scenario through a weighted combination of several components, and transform this utiliy to a probability of help ($P_{Help}$) through a softmax function:

$$P_{Help} = \frac{1}{1 + e^{-\beta \cdot U}} \quad (1)$$

$\beta$ is the inverse temperature parameter that controls the randomness of the decisions. $\beta \in (0, 10]$.

According to existing studies, there are several possibilities regarding how the $U$ is calculated (e.g., dependent vs. independent weights, linear vs. nonlinear combinations, etc.[8,11]). Therefore, we tested and compared among a total of seven model candidates with increasing complexity while calculating $U$. In the first step, we compared among three linear models (1.0, 2.0, 3.0) that differed in what variables they included and how the variables were combined through weights.

**1.0 Fixed helping bias model**: In the fixed helping bias model, the utility of helping is entirely driven by the disposition of helping behavior, represented by a parameter ($b$). Thus one would have a fixed value/probability of helping regardless of the scenarios.

$$U = b \quad (2)$$

**2.0 Single weight cost-benefit model**: In the single weight cost-benefit model, the utility of helping is calculated not only by the disposition but also based on the cost and benefit of the helping action in the actual scenarios. Here, benefit and cost are defined by the min-max normalized Factor 1 (both benefit) and Factor 2 (self-cost) scores, respectively. We performed the min-max normalization to restrict the cost and benefit to [0,1], for the convenience of modeling. The cost and benefit are combined linearly through one weight parameter ($w_{cost}$):

$$U = -w_{cost} \cdot Cost + (1 - w_{cost}) \cdot Benefit + b \quad (3)$$

Note that for both $Cost$ and $Benefit$ variables, the higher the value, the bigger the cost/benefit. We restricted $w_{cost}$ so that $w_{cost} \in [0, 1]$.

**3.0 Double weight cost-benefit model**: Compared to the single weight model, here the cost and benefit are combined through two independent weights, $w_{cost}$ and $w_{benefit}$:

$$U = -w_{cost} \cdot Cost + w_{benefit} \cdot Benefit + b \quad (4)$$

Similar to 2.0, $w_{cost}, w_{benefit} \in [0, 1]$.

A comparison among the above three models revealed that model 2.0 fit the best. Therefore, we explored model 2.0 with more variations by introducing non-linearity (2.1, 2.2, 2.3) and adding another independent urgency term (2.4).

**2.1 Nonlinear cost model**: In the nonlinear cost model, the cost is combined with benefit after a nonlinear transformation, characterized by an additional parameter $\alpha$:

$$U = -w_{cost} \cdot Cost^{\alpha} + (1 - w_{cost}) \cdot Benefit + b \quad (5)$$

**2.2 Nonlinear benefit model**: In the nonlinear benefit model, the benefit undergoes a nonlinear transformation characterized by $\alpha$:

$$U = -w_{cost} \cdot Cost + (1 - w_{cost}) \cdot Benefit^{\alpha} + b \quad (6)$$

**2.3 Interactive model**: In the interactive model, an interaction term $Cost \cdot Benefit$ with weight $w_{cb}$ is added in addition to the linear combination of cost and benefit:

$$U = -w_{cost} \cdot Cost + (1 - w_{cost}) \cdot Benefit + w_{cb} \cdot Cost \cdot Benefit + b \quad (7)$$

**2.4 Urgency bonus model**: Existing studies have found that people are more likely to help others under emergency situations compared to non-emergencies[48–50]. Therefore, we derived the urgency bonus model, in which the urgency of the situation modulates the utility of helping through a bonus mechanism. We binarized the consensus rating of the situational urgency level ("How urgent is this situation?") to 0 and 1 by a median split. The utility of help is then computed as the sum of cost, benefit, as well as an urgency bonus $w_u$ under a high urgency situation:

$$U = -w_{cost} \cdot Cost + (1 - w_{cost}) \cdot Benefit + w_u \cdot Urgency + b \quad (8)$$

Here, $Urgency = 0$ in low urgency scenarios, $Urgency = 1$ in high urgency scenarios.

## Model fitting, model comparison, and parameter recovery

We fit the models to each participant's data with the maximum likelihood estimation through the 'SciPy' package (version 1.13.1) and custom Python scripts. Based on the model fits, we calculated several model diagnostics for model comparison, including the pseudo $R^2$, Akaike Information Criterion (AIC), Bayesian Information Criterion (BIC), and out-of-sample prediction accuracy. The out-of-sample prediction accuracy was calculated using the average leave-one-out prediction accuracy (across all the trials).

We then conducted a parameter recovery analysis by simulating 100 datasets with 100 sets of simulated parameters and fit the simulated data. This procedure was repeated 10 times. We then compared the fitted parameters (i.e., 'recovered parameters') with the simulated parameters using Pearson's correlations. A parameter is considered to be well recovered if the correlation is high.

## Model robustness check

Although our primary focus is to model the help vs. not help decision as a binary variable, we additionally replicated the selected model (2.0) with a continuous rating version to check the robustness of the model fitting results. This continuous version (hereafter named 'rating model') was coded the same way as the binary version (hereafter named 'decision model'), except that (1) we applied a scaling factor to scale up the estimated utility to match the range of WTH rating rather than calculating a probability with the softmax function, and (2) we optimized by minimizing the mean squared error (MSE) between the predicted rating and actual ratings.

We compared the rating model with the decision model in multiple aspects. First, to check whether the rating model can obtain similar model fitting results, we correlated the model performance metric (MSE for the rating model, and log likelihood for the decision model) and estimated parameters ($w_{cost}$, $b$) between the two models. Second, we checked whether the rating model showed similar predictive properties by calculating (1) the out-of-sample predictions of WTH rating, (2) correlation between $w_{cost}$ and WTH ratings under high vs. low cost conditions, (3) correlation between $w_{cost}$ and WTH ratings under high vs. low benefit conditions, and (4) correlation between $b$ and WTH. All the above analyses confirmed that the rating model showed highly consistent results with the decision model, suggesting high robustness of the original modeling approach.

## Representational space of the helping narratives

The everyday helping scenario set could serve as a general research resource for future studies. To better understand the properties of these scenarios, we characterized them under three representational spaces using three sources of information: the population-level willingness to help (decision space), the cost and benefit (motivation space), and the semantic features of the narratives (semantic space).

**Decision space.** The decision space was constructed based on the (dis) similarities of average WTH (across all participants) among all pairs of scenarios. We first defined the dissimilarity $F_{decision}(S_x, S_y)$ between any two scenarios $S_x$ and $S_y$, as the average WTH rating differences (both signed and unsigned/absolute) difference between them across raters:

$$F_{decision}(S_x, S_y) = \frac{1}{n}\left[ 0.5 \cdot \sum_{i=1}^{n} |WTH_{i,x} - WTH_{i,y}| \right. \\ \left. + 0.5 \cdot |\sum_{i=1}^{n}(WTH_{i,x} - WTH_{i,y})| \right] \tag{9}$$

$WTH_{i,x}$ represents rater i's WTH rating on scenario $S_x$, $n$ represents the total number of raters who rated both $S_x$ and $S_y$. In the unsigned difference term, two scenarios are similar to each other if every rater rated them consistently (i.e., either both high or both low), and two scenarios are dissimilar if raters rated them in divergent directions (i.e., rate $S_x$ very high but $S_y$ very low). In the signed difference term, two scenarios are similar if their average WTH ratings are similar. With the (dis)similarity measure, we constructed a pairwise dissimilarity matrix across all scenarios and performed multi-dimensional scaling (MDS) based on the dissimilarity matrix ('MDS' function from scikit-learn version 1.5.0, Python version 3.12.3). MDS produced a 2-dimensional representation of the scenarios.

**Motivation space.** A 2-dimensional motivation space was constructed using the two factors (both benefit, self cost) identified from the EFA. 100 scenarios were projected to this space by assigning the (x,y) coordinates with (both benefit, self cost). The space was then split to four quadrants using the medians of the factor scores, and the quadrants represented four categories of scenarios: 1) low benefit, low cost (23 scenarios), 2) low benefit, high cost (27 scenarios), 3) high benefit, low cost (27 scenarios), and 4) high benefit, high cost (23 scenarios). For subsequent analyses, we defined the pairwise dissimilarity in the motivation space $F_{motivation}(S_x, S_y)$ to be the Euclidean distance between any two scenarios.

**Semantic space.** We performed text embedding for every scenario narrative using the OpenAI 'text-embedding-3-large' model, and obtained the embedding as vectors in a 256-dimensional space. As the text-embedding space was extremely high-dimensional, we performed a dimensionality reduction using Uniform Manifold Approximation and Projection (UMAP), and visualized the reduced space on a 2D plane, which became a semantic space of scenarios ('UMAP' function from the 'umap-learn' package, version 0.5.6). During the UMAP, we defined the pairwise distance using the cosine distance between embedded vectors of the scenarios. The pairwise dissimilarity in the semantic space $F_{semantic}(S_x, S_y)$ was defined as the Euclidean distance between scenarios. We categorized the scenarios into 14 semantic categories using a combination of data-driven clustering and human interpretation: an initial clustering was generated using the agglomerative hierarchical clustering ('SciPy' package, version 1.13.1), resulting in 12 data-driven clusters; next, two human annotators (Q.W. and M.S.) independently adjusted the labels so that each cluster could be summarized with a highly interpretable common topic; the human annotators then discussed and reached a consensus on the final set of labels and categories.

## Associations among representational spaces

To understand how WTH may be affected by the 2 motivational factors (cost and benefit), we compared the WTH across four quadrants in the motivational space using Welch's one-way ANOVA (as the variables had unequal variances tested with a Bartlett's test) and the Games-Howell post-hoc pairwise comparisons.

To test whether the three spaces were inter-related to each other, We performed representational similarity analyses between every pair of spaces. While constructing the latent spaces, we obtained three 100*100 representational dissimilarity matrices (RDM). Each element $RDM(S_x, S_y)$ was the dissimilarity between scenario $x$ and scenario $y$ under the definition of each space (i.e., $F_{decision}(S_x, S_y)$, $F_{motivation}(S_x, S_y)$, $F_{semantic}(S_x, S_y)$). We next calculated the Pearson's correlations between all possible pairs of the RDMs (after converting each RDM to an array by taking the elements from the upper triangle). Finally, permutation tests were performed to calculate the statistical significance of the correlations: the same procedure was repeated 5000 times while randomly shuffling the row and column orders of one RDM in order to derive a null distribution of the correlations. p-value was determined as the proportion of permutations whose resulting correlation is higher than the actual correlation.

In addition to the correlation among spaces, we also asked whether the variability of the motivations and semantic meanings across the scenarios could explain the variability of helping decisions. To answer this question, we performed two multiple linear regression analyses to fit the decision space variability (i.e., RDM) with the variability of other spaces. The decision space RDM was decomposed to two components: a WTH RDM based on the absolute difference of mean WTH ratings between pairs of scenarios, and a residual RDM by regressing the WTH RDM out of the decision space RDM ($RDM_{decision} = RDM_{WTH} + residual$). Meanwhile, the motivation space RDM was decomposed to a benefit RDM (pair-wise absolute differences of the mean benefit factor scores) and a cost RDM (pair-wise absolute differences of the mean cost factor scores). In the first regression model, we predicted the WTH RDM using the benefit RDM, cost RDM, and the semantic space RDM ($RDM_{WTH} = RDM_{benefit} + RDM_{cost} + RDM_{semantic}$). In the second regression model, we predicted the decision space residual RDM using the benefit RDM, cost RDM, and the semantic space RDM ($residual = RDM_{benefit} + RDM_{cost} + RDM_{semantic}$). The regressions were performed using the 'statsmodels' package (version 0.14.4) in Python.

## Individual differences in helping decisions

A set of statistical tests was conducted to understand how individual differences in helping behavior identified through the behavioral data and computational models were associated with various demographic characteristics and psychological traits.

First, we explored what demographic variables could explain the overall WTH across individuals. We performed Welch's independent sample t-tests (two-sided) on WTH between two sex groups (male vs. female) and between two ethnicity groups (Hispanic/Latino vs. non-Hispanic/Latino). Then we performed Welch's one-way analysis of variance (ANOVA) on WTH across three gender groups (male, female, non-binary), five racial groups (Black/African American, white, Asian, Native American, multiracial), six educational levels (some high school or less, high school diploma or GED, Associates or technical degree, some college, Bachelor's degree, Graduate degree), and six income levels (<$25,000, $25,000 - 49,999,$50,000 - 74,999, $75,000 - 99,999, $100,000 - 149,000, > $150,000), respectively. Finally, we calculated the Pearson's correlation between age and WTH. Since we tested several variables, we corrected the p-value for multiple comparisons using the Bonferroni correction. For both the t-test and ANOVA, we tested the normality and equal variances assumptions using the D'Agostino and Pearson's omnibus test of normality, Levene test for equality of variances (two samples), and Bartlett's test for homogeneity of variance (>two samples).

Second, we explored if any psychological trait is associated with the WTH. We computed the Pearson's correlation between WTH and each of the five candidate traits: dispositional empathy (measured by the empathic

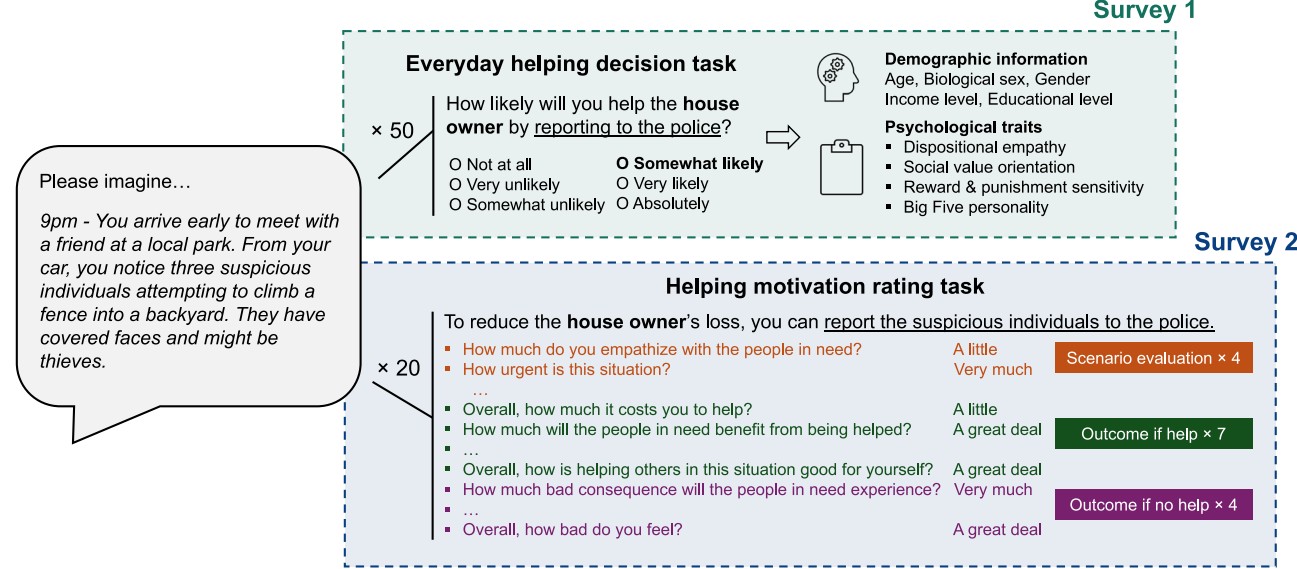

**Fig. 1 | Experimental procedure and the task flow.** A total of $N_1 = 215$ participants completed Survey 1, and a subset of them ($N_2 = 140$) completed Survey 2 as well. In both surveys, participants were presented with brief scenario narratives in which one or more people are in need of help. In the Survey 1 everyday helping decision task, participants were provided with a helping action and rated their willingness to help for 50 scenarios. In the Survey 2 helping motivation rating task, participants evaluated their motivations to help (or not help) in 15 different aspects for 20 scenarios. In addition, all participants reported their demographic information and answered a set of personality trait questionnaires at the end of Survey 1.

concern subscale of IRI), social value orientation (measured by SVO), reward responsiveness (measured by the RR), punishment sensitivity (measured by BIS), and agreeableness (measured by the agreeableness subscale of BFI-10). The Bonferroni correction was also applied to adjust for multiple comparisons.

Next, we tested the correlations between psychological traits and the fitted model parameters from model 2.0, including $w_{cost}$ and $b$. We hypothesized that the weight of cost relative to benefit ($w_{cost}$) positively correlates with punishment sensitivity and negatively correlates with reward responsiveness; and that the dispositional measure of one's helping tendency ($b$) positively correlates with both dispositional empathy and agreeableness. Therefore, we calculated Pearson's correlations among the aforementioned pairs of variables.

In addition, we explored how different traits (questionnaire-assessed and model-derived) influence helping behavior across different categories of scenarios. For each trait, we divided the participants into a high trait group and a low trait group through a median split. We then quantified the WTH differences of each semantic category of scenarios between high trait and low trait groups through Cohen's d. According to existing criteria[51], a d (absolute value) of 0.2 is considered small, 0.5 is medium, and 0.8 is large. A positive d means individuals higher in a trait are more likely to help; and a negative d means individuals lower in a trait are more likely to help.

All the statistical comparisons were conducted using the 'SciPy' (version 1.13.1) and 'Pingouin' (version 0.5.5) package in Python (version 3.12.3). Note that we used the permutation method to calculate p values of the Pearson's correlation, and the bootstrapping method to generate confidence intervals (both methods do not have restrictions on the data distributions).

**Content analysis for additional self-reported motivations**
During Survey 2, in the concluding question of each rated helping scenario, participants provided open-ended responses describing their dominant motivations in their helping decision-making processes. The objective was to explore and identify potential additional motivations not accounted for by the predefined rated dimensions.

We conducted a content analysis to extract all self-reported motivations, regardless of whether they reflected positive or negative influences on helping behavior. Motivations not included in the study's initial set of rated dimensions were retained for further examination.

Two independent human annotators (Q.W. and M.S.) labeled and classified these additional motivations, organizing them into highly interpretable categories. After independent coding, the annotators discussed and reached consensus on a final set of motivation categories and their corresponding definitions. Subsequently, basic descriptive statistics were computed to summarize the distribution of additional motivations.

## Results
### Evaluations on everyday helping scenarios
We developed a text-based stimulus set consisting of 100 second-person perspective, short descriptions of everyday life situations in which one or more people are in need of help ("everyday helping scenarios stimulus set", Supplementary Data 1). In Survey 1, $N_1 = 215$ participants (110 females, 105 males) completed an everyday helping decision task ('decision task' hereafter), in which they rated 50 different helping scenarios on their willingness to help (WTH) from -5 to 5, and subsequently answered a set of demographic and personality questionnaires. In Survey 2, a subset of participants ($N_2 = 140$, 71 females, 68 males, 1 unknown) also completed a helping motivation rating task ('rating task' hereafter) and each rated 15 motivation-related features (Supplementary Table 1) for 20 scenarios (see experimental procedures in Fig. 1 and recruitment in Supplementary Fig. 1). The two samples were matched in demographic profiles and scenario ratings (Table 2). According to the WTH ratings from the decision task, these scenarios represent great heterogeneity as to whether the general population would like to offer help (Fig. 2a): the WTH ratings ranged from −2.31 to 4.24 (Mean = 1.52, s.d. = 1.62), and the percentage of helping decisions (i.e., binary using zero as a cut-off) ranged from 21.90% to 99.04% (Mean = 72.29%, s.d. = 19.84%) across all scenarios. That said, participants overall tended to help (Fig. 2b): the average WTH across participants was 1.52 (s.d. = 1.35), and the average percentage of help decisions was 72.19% (s.d. = 17.56%).

Aggregating participant responses from the rating task, we derived group-level consensus ratings on the motivation-related features for each scenario by calculating the average ratings among responses that passed a set of quality checks (see details in "Methods"). For all the features, the

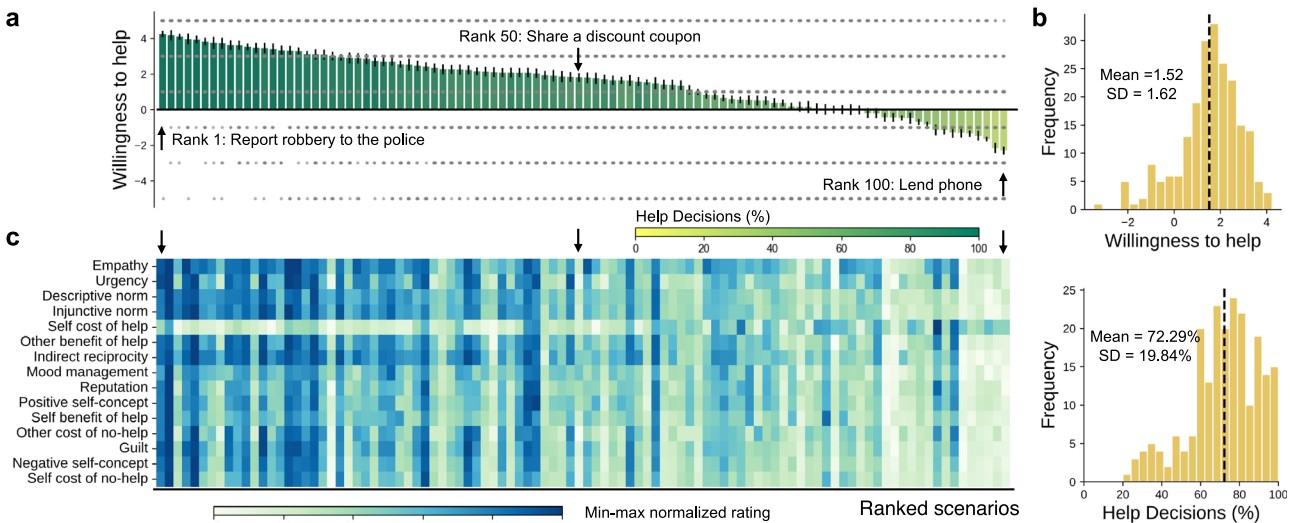

**Fig. 2 | Overview of the scenario ratings. a** Willingness to help among 100 scenarios. Bar plots represent the average willingness to help (WTH) among 100 scenarios, arranged in a descending order of the WTH. Gray dots show distributions of the ratings per scenario from N = 215 individual participants. A WTH of −5 means "Not likely at all", and a WTH of 5 means "Absolutely". Error bars represent standard errors. The bars are colored by the help decisions (i.e., percentage of help after binarizing the WTH at a 0 cutoff). Three example scenarios at the 1st, 50th, and 100th rank are highlighted. **b** Distribution of WTH rating (top) and frequency of help decisions (bottom) across N = 215 participants. Dotted lines denote the mean. **c** Consensus rating of 15 motivation dimensions among 100 scenarios. Each cell in the heatmap represents the average rating of one motivation dimension in one scenario, after min-max normalization across all scenarios. Each column represents one scenario, and the scenarios are arranged in the same order as in **a**. Darker blue indicates higher rating, lighter blue indicates lower rating.

consensus ratings showed substantial variations from scenario to scenario (Fig. 2c) and maintained good to excellent inter-rater reliability as assessed with the intra-class correlation coefficient (ICC; ICC range: [0.822, 0.966], Supplementary Table 1). Note that we intentionally recruited raters as a subset of Survey 1 participants so that the ratings could well represent the evaluations from sample 1 during the subsequent decision modeling (i.e., not biased by any potential difference between the two samples).

We assessed the predictive performance of each motivational feature to the WTH ratings, and found that all hypothesized motivation dimensions explained a medium to large portion of WTH variance across scenarios and individuals. Despite an overall good performance, variability existed: among all motivations, injunctive norm (peers think one should help) explained most variance (marginal $R^2 = 0.358$, conditional $R^2 = 0.817$, derived from a linear mixed effect model), and other's cost of no-help explained the least variance (marginal $R^2 = 0.214$, conditional $R^2 = 0.490$, Supplementary Fig. 2).

**Computational models of helping decisions**

Our first goal was to establish a computational model that quantitatively characterized the relationships between helping decisions and their underlying motivations. As many of the motivation dimensions collected through the rating task were highly correlated (Supplementary Fig. 3), they might be sufficiently represented by a smaller set of latent dimensions. Therefore, we performed an exploratory factor analysis (EFA) on 12 motivation dimensions (empathy, descriptive norm, injunctive norm, self cost, other benefit, indirect reciprocity, mood management, reputation, positive self-concept, other cost, guilt, negative self-concept) and successfully identified 2 orthogonal factors (via varimax rotation) that accounted for 87% of total variance (Supplementary Fig. 3). The first factor comprised 11 motivation dimensions (i.e., all but self cost) that are relevant to a positive driving force of help. Because most questions asked about the benefit of help to both the helper and the helpee, we named this factor "Both Benefit" (or benefit). The second factor was dominated by the self cost, representing a driving force of not helping; thus we named this factor "Self Cost" (or cost).

We next examined how the "Both Benefit" and "Self Cost" factors influence one's willingness to help. Previous studies have proposed several

models in which the utility of prosocial actions was computed through weighted combinations of cost and benefit. These models differ in whether the weights on cost and benefit are dependent (2.0) or independent (3.0), and whether the combination is linear (2.0, 3.0) or nonlinear (2.1, 2.2: exponential scaling of a factor, 2.3: interaction between cost and benefit, 2.4: modulation by additional factors). As such, we tested a total of 7 models to fit the binarized help decisions of each participant (Fig. 3a). Among all model candidates, the single weight cost-benefit model (model 2.0) performed the best (lowest AIC, BIC, highest out-of-sample accuracy, Fig. 3, Supplementary Table 2). All the parameters in this model were well recovered and we obtained similar modeling results when predicting the rating scores instead of binary decisions (Supplementary Fig. 4, 5).

In the single weight cost-benefit model, the utility of help is computed by a linear combination of the cost and benefit using a single weight $w_{cost}$, as well as an additional term of helping bias $b$:

$$U = -w_{cost} \cdot Cost + (1 - w_{cost}) \cdot Benefit + b$$

Here, $w_{cost}$ represents the relative importance of cost over benefit when making the decision. It is positively associated with the WTH differences between low cost and high cost conditions (Pearson's $r$ (213) = 0.432, $p < 0.001$, 95%CI = [0.328, 0.526], Fig. 3c), and negatively associated with the WTH differences between high benefit and low benefit conditions (Pearson's $r$ (213) = −0.204, $p = 0.002$, 95%CI = [−0.331, −0.059], Fig. 3d). In other words, one who cares a lot about cost (high $w_{cost}$) has a greater WTH decrease in high cost compared to low cost scenarios; and one who cares a lot about benefit (low $w_{cost}$) has a greater WTH increase in high benefit compared to low benefit scenarios. Meanwhile, $b$ represents a predisposition to help regardless of the cost and benefit of the situation, and is positively correlated with the average WTH (Pearson's r(213) = 0.522, $p < 0.001$, 95%CI = [0.421, 0.624], Fig. 3e).

Our model successfully reproduced the key features of the dataset: when using it to predict (i.e., leave-one-out, out-of-sample prediction) the average percentage of help decisions across different scenarios, the predicted percentage was highly correlated with the actual data (Pearson's $r$ (98) = 0.824, $p < 0.001$, 95%CI = [0.748, 0.878], Fig. 3f).

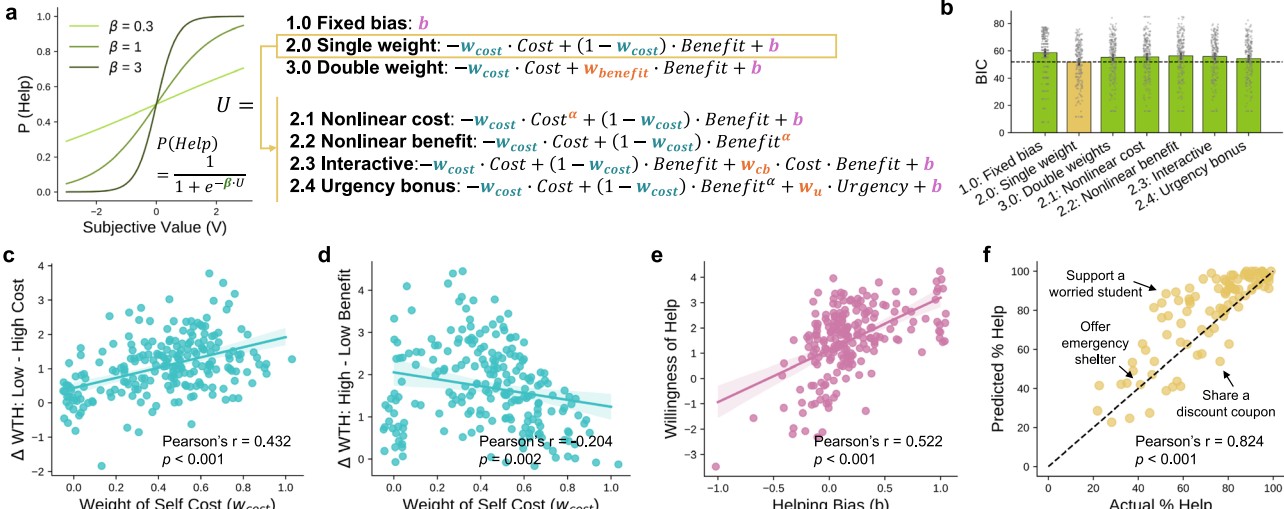

**Fig. 3 | Computational modeling of helping decisions. a** Overview of the model candidates. In all the models, the utility of helping is first computed, and go through a softmax function to determine the likelihood of a help decision. The shape of the softmax function is controlled by a `beta` parameter. To compute the utility, we first compared among three classes of models (1.0, 2.0, 3.0), and found that the single weight model (i.e., the weights of cost and benefit sum to 1) performed the best. Next, we derived four additional variations based on the single weight assumption, that featured different types of nonlinearity (2.1, 2.2, 2.3) and an urgency bonus (2.4). **b** Model comparison. Across all 7 models, the 2.0 single weight model had the smallest BIC. Error bars represent standard errors. **c** Correlation between weight of self cost parameter and WTH rating differences under low vs. high cost scenarios. Each dot represents a participant. **d** Correlation between weight of self-cost parameter and WTH rating differences under high vs. low benefit scenarios. Each dot represents a participant. **e** Correlation between helping bias parameter and WTH. Each dot represents a participant. **f** Out-of-sample prediction of scenario-wise average WTH. Each dot represents a scenario. The actual percentage of helping decision was highly correlated with the predicted percentage of helping decision across all scenarios. We highlighted three example scenarios whose actual percentage of helping decision was above, below, or equal to the prediction. **b–e** display data from $N = 215$ participants, **f** displays data from $N = 100$ scenarios.

## Individual differences in helping decisions are related to various personality dimensions

In accordance with our second goal, the decision task and the rating task collectively showcased the idiosyncrasies of helping behavior among the general population through a wide distribution of both the WTH ratings and the model parameters. We then asked whether individual differences in helping behavior are associated with demographic characteristics and personality traits.

We first compared the overall WTH level across groups defined by several demographic characteristics, including biological sex, gender, age, race, ethnicity, educational level, and income level (Supplementary Fig. 6). However, there were no statistically significant associations between WTH and any of the demographic variables.

Next, we explored the correlations between WTH and five personality traits that may be relevant (Fig. 4a), including dispositional empathy (IRI-EC), social value orientation (SVO), reward responsiveness (RR), punishment sensitivity (BIS), and agreeableness (BFI-A). Among the five traits, we found significant positive correlations (after Bonferroni correction) between WTH and dispositional empathy (Pearson's $r(213) = 0.476$, $p = 0.001$, 95% CI = [0.342, 0.596]), reward responsiveness (Pearson's $r(213) = 0.212$, $p = 0.010$, 95% CI = [0.069, 0.338]), and agreeableness (Pearson's $r(213) = 0.329$, $p = 0.001$, 95% CI = [0.202, 0.447]).

Finally, we tested several hypotheses about which personality traits are relevant to the model parameters: weight of cost ($w_{cost}$) and helping bias ($b$). Because the weight of cost controls to what extent people value cost relative to benefit, we hypothesized that people with a higher $w_{cost}$ are more sensitive to situations that involve bad consequences (or punishment), and less active in response to rewards. Correlation analyses affirmed the former but not the latter: $w_{cost}$ was positively correlated with punishment sensitivity (Pearson's $r(213) = 0.163$, $p = 0.016$, 95% CI = [0.035, 0.290], Fig. 4b), yet there was no statistically significant correlation with the reward responsiveness (Pearson's $r(213) = 0.007$, $p = 0.929$, 95% CI = [−0.130, 0.146], Fig. 4c). In addition, we hypothesized that people with a higher bias toward helping ($b$) are more empathetic, and have more agreeable personalities. Indeed, significant positive correlations were found between $b$ and both dispositional

empathy (Pearson's $r(213) = 0.159$, $p = 0.018$, 95% CI = [0.017, 0.316], Fig. 4d) and agreeableness (Pearson's $r(213) = 0.183$, $p = 0.005$, 95% CI = [0.057, 0.312], Fig. 4e).

## Characterizing 100 helping scenarios in the decision space, motivation space, and semantic space

Having elucidated the computational mechanism of helping decision-making and the personality correlates of its heterogeneity, our last goal was to comprehensively characterize the helping scenario stimulus set to facilitate its use in future research. To achieve this goal, we established latent representations of these scenarios using three independent approaches and visualized the scenarios in three latent spaces: a decision space, a motivation space, and a semantic space (Fig. 5).

The decision space reflects the outcomes of a helping scenario, in which scenarios with similar WTH ratings across participants are located close to each other (Fig. 5a). Therefore, as expected, the main source of variation in the decision space comes from the average WTH ratings across scenarios (Fig. 5b). Nevertheless, among scenarios that have the same average WTH rating, one pair that was consistently rated at the same level (e.g., 3, 3, 3 for rater 1, 2, 3, respectively) would have a smaller distance than another pair that was rated not consistently across participants (e.g., 1, 3, 5, for rater 1, 2, 3, respectively).

The motivation space was defined by the two motivational dimensions (benefit, cost) identified from the factor analysis. (Fig. 5c, d). Along both dimensions, scenarios are widely distributed, and were categorized into four quadrants (split by the median of each dimension): low benefit–high cost (27 scenarios), high benefit–high cost (23 scenarios), low benefit–low cost (23 scenarios), and high benefit–low cost (27 scenarios).

The semantic space reflects the similarity of the narrative language among the scenarios, regardless of any subjective human evaluation. The scenario narratives went through large language model-based text embeddings and were further projected to a 2-dimensional space for visualization. Based on the semantic meaning of the scenarios and their locations in the space, we assigned the scenarios to 14 semantic groups that featured different topics/key

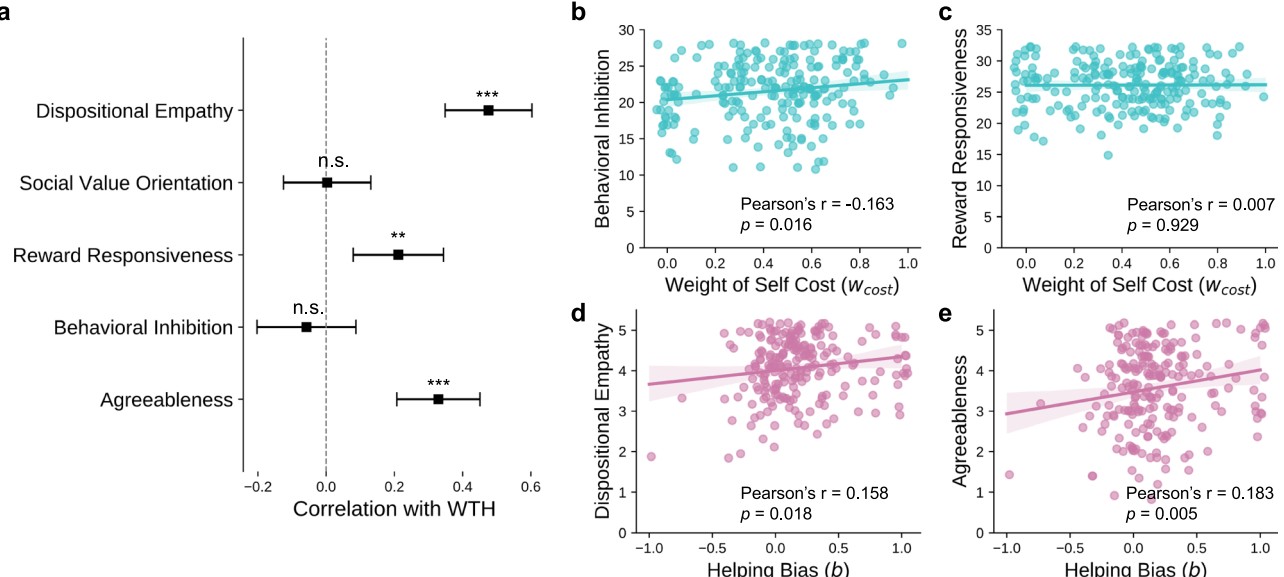

**Fig. 4 | Individual differences in helping behavior are associated with psychological traits. a** Correlations between mean WTH rating and five psychological traits. Points represent the Pearson's correlation, error bars represent the 95% confidence interval with bootstrapping methods. n.s. not significant, *$p < 0.05$, **$p < 0.01$, ***$p < 0.001$, all after Bonferroni correction. **b** Correlation between $w_{cost}$ and the behavioral inhibition score (measured by BIS). **c** Correlation between $w_{cost}$ and the reward responsiveness (measured by RR). **d** Correlation between $b$ and dispositional empathy (measured by IRI-EC). **e** Correlation between $b$ and agreeableness (measured by BFI-A). All the correlations are calculated across $N = 215$ participants.

words: safety (14 scenarios), commercial area (11 scenarios), street encounter (9 scenarios), community welfare (9 scenarios), crime (8 scenarios), (heavy) belongings (8 scenarios), searching (7 scenarios), young generation (6 scenarios), social media support (6 scenarios), mobility prosociality (6 scenarios), vulnerable group (5 scenarios), pet care and issues (4 scenarios), community issues (4 scenarios), and act of God (3 scenarios) (Fig. 5f, see a full list of labels in Supplementary Data 1).

**Associations among three latent scenario spaces**

After defining and visualizing the three spaces, we asked whether two scenarios close to each other in one space are also similar in other spaces. In other words, whether the similarity representations revealed any shared information among these three independently constructed spaces. Using a representational similarity analysis (RSA), we found that the pairwise similarities across scenarios are correlated in all three spaces (decision space vs. motivation space: Pearson's $r(4948) = 0.485, p < 0.001$; decision space vs. semantic space: Pearson's $r(4948) = 0.122, p < 0.001$; motivation space vs. semantic space: Pearson's $r(4948) = 0.217, p < 0.001$, p-values were based on permutation tests, Fig. 5), suggesting shared similarity structures among them.

To further understand the associations between spaces, we visualized their relationships in Fig. 6. Across the four quadrants in the motivation space, participants were more willing to help in the low cost - high benefit scenarios ($n = 27$, Mean = 2.77, s.d. = 0.80), followed by the high cost–high benefit ($n = 23$, Mean = 1.96, s.d. = 1.68), low cost–low benefit ($n = 23$, Mean = 1.37, s.d. = 1.33) and finally high cost–low benefit ($n = 27$, Mean = 0.026, s.d. = 1.15) scenarios (Fig. 6a). A one-way ANOVA revealed significant differences across the four motivation quadrants (F(3, 49.5) = 34.65, $p < 0.001$, $\eta_p^2$=0.414, 95%CI = [0.176,0.547]). The subsequent post-hoc Games-Howell test showed that the high cost–low benefit quadrant had significantly smaller WTH than all the other quadrants, and that the low cost–high benefit quadrant had significantly higher WTH than the low cost–low benefit quadrant. We did not find statistically significant differences between the high cost–high benefit quadrant and low cost–low benefit/low cost–high benefit quadrants (see all the statistics in Supplementary Table 3). Overlaying the benefit and cost scores onto the decision space also

indicated correlated patterns between the WTH and benefit/cost (Supplementary Fig. 7). After calculating the correlations between these variables, we found a significant positive correlation between the WTH and benefit (Pearson's $r(98) = 0.723, p < 0.001$, 95%CI = [0.611, 0.800]) and a negative correlation with cost (Pearson's $r(98) = -0.469, p < 0.001$, 95%CI = [-0.607, -0.303]).

Across all the semantic categories, there are variations regarding the WTH and benefit/cost ratings. Figure 6b illustrates the average WTH levels among all semantic categories in descending order, while Fig. 6c shows the average benefit/cost corresponding to each of the categories. The highest WTH falls in the (heavy) belongings category (e.g., lift a heavy bag), featuring a moderate level of benefit yet a small amount of cost; it is followed by the vulnerable group (e.g., give a seat to a pregnant woman), crime (e.g., stop harassment), and safety (e.g., call ambulance for someone) categories, which all have high levels of benefit and a medium level of cost. Two categories that have the smallest WTH are community issues (e.g., report apartment maintenance) and community welfare (e.g., draft a letter to government), both showing small to medium levels of benefit but medium to high levels of cost.

The above results indicate that different benefit and cost profiles in different semantic categories are associated with the differences in overall WTH across the scenarios. However, even when two scenarios are judged with the same average WTH, they may not be evaluated in the same way by each individual (see further explanation in Supplementary Fig. 8). What accounts for this within-rater consistency? To answer this question, we partitioned the scenario decision similarities into two components by decomposing the decision space representational dissimilarity matrix (RDM) into a WTH RDM (i.e., an RDM calculated based on the mean WTH rating differences) and a residual RDM (that accounts for additional variances due to within-rater variability). We found that although the WTH RDM was significantly predicted by benefit ($\beta = 0.476$, s.e. = 0.012, $p < 0.001$, 95% CI = [0.452, 0.501]) and cost ($\beta = 0.229$, s.e. = 0.013, $p < 0.001$, 95% CI = [0.205, 0.254]) RDM (but not the semantic space RDM) the residual RDM was significantly predicted by the semantic space RDM alone ($\beta = 0.070$, s.e. = 0.015, $p < 0.001$, 95% CI = [0.041, 0.098]). These results suggested that although the average group consensus of WTH level is associated with the costs and benefits, scenarios from the same semantic categories are

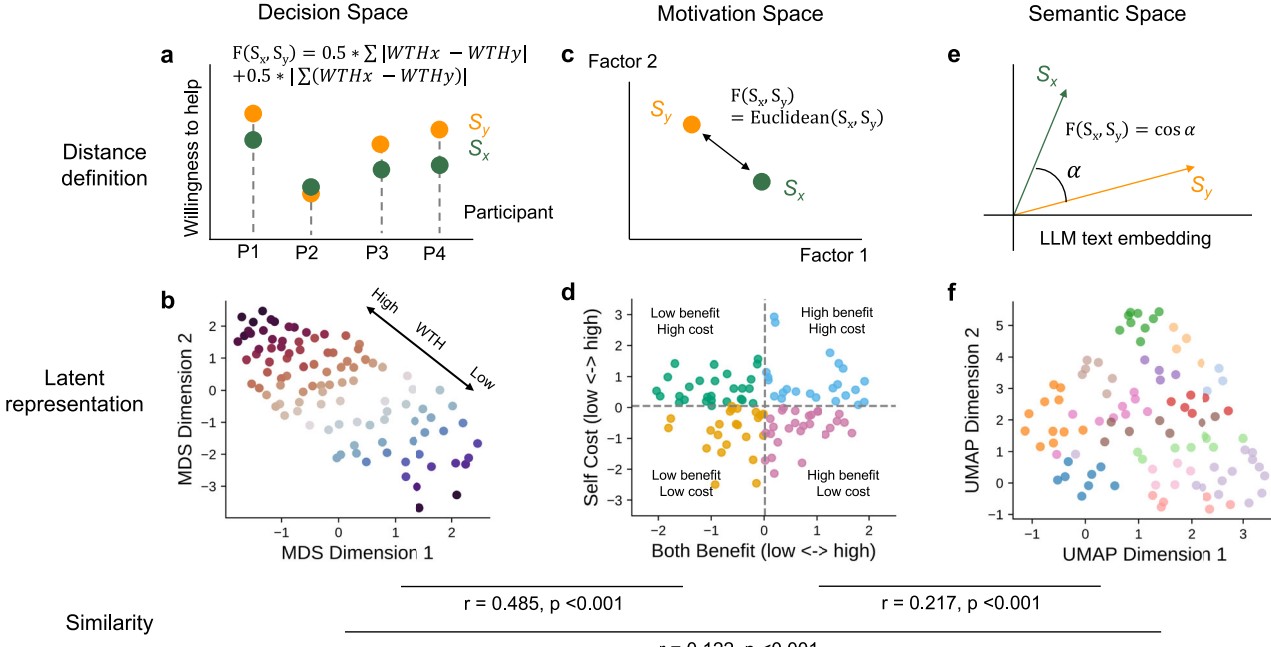

**Fig. 5 | Latent representation of 100 scenarios in three spaces.** Top row: illustration of the space construction methods. **a** Decision space. Decision space dissimilarity was defined by the cross-participant variation of the WTH between every pair of scenarios (sum of both signed and unsigned/absolute difference across all participants). **c** Motivation space. Motivation space has two dimensions corresponding to Factor 1 (Both benefit) and Factor 2 (Self cost). The dissimilarity in the motivation space was defined as the Euclidean distance. **e** Semantic space. Semantic space was constructed by UMAP mappings from the 256-d LLM text embedding vectors of each scenario to a 2D space. Green dots and arrows represent scenario $x$, orange dots and arrows represent scenario $y$. Middle row: visualization of the latent representations of 100 scenarios in **b** Decision space (color coded by average WTH, red indicates positive and blue indicates negative values), **d** Motivation space (color coded by four quadrants after median splits), and **f** Semantic space (color coded by semantic labels). Bottom row: representational similarity analysis among pairs of spaces. $R$ is the Pearson's correlation between the representational dissimilarity matrices (RDM) of one space and the other. $p$ indicates the statistical significance of the correlation using a permutation approach.

evaluated more consistently within an individual. In other words, individuals tend to use consistent criteria when judging scenarios of the same semantic type.

## Personal traits differentially modulate helping in different types of scenarios

As we obtained meaningful semantic categories of the helping scenarios, we further asked whether personal traits influence helping behavior in a context-dependent manner—that is, whether some of the traits selectively impact WTH in some types of helping situations but not others. We investigated this nuanced relationship by comparing the differences in WTH ratings (of each semantic category) between individuals high in and low in each trait (including five questionnaire-assessed personality traits and two model-based parameters, Fig. 7). Consistent with our previous analyses, the helping bias, dispositional empathy, agreeableness, and reward responsiveness showed medium effect sizes on most scenario categories, confirming their role as general indicators of helping intentions. Beyond such a general effect of traits, we also observed context dependency between traits and scenarios: a given trait could have larger effects on a subset of scenarios than others; and WTH of a given scenario category could be better explained by a few traits.

More specifically, the two model parameters showed complementary effects among scenario types: while the helping bias showed greater effects for predicting most types of scenarios (Cohen's $d$ range: 0.16 to 1), the weight of self cost is more predictive of helping in the 'vulnerable group' (Cohen's $d = -0.44$ 95% CI = [$-0.71, -0.17$]), 'pet care and issues' (Cohen's $d = -0.38$ 95% CI = [$-0.65, -0.11$]) and 'safety' (Cohen's $d = -0.35$, 95% CI = [$-0.62, -0.08$]) scenarios—scenarios that typically have high benefit and high cost. Similarly, across the five personality traits, dispositional empathy showed greater effects in most scenarios (Cohen's $d$ range: 0.18 to 0.70), yet behavioral inhibition had a stronger association with "community

issues" (Cohen's $d = -0.37$, 95% CI = [$-0.64, -0.10$]) scenarios, which feature high cost and low benefit. Social value orientation, on the other hand, showed minimal effects regardless of the context (Cohen's $d$ range: $-0.16$ to 0.17).

## Additional self-report motivations

Finally, to understand additional motivations for helping decisions that were not accounted for by the rated dimensions, we collected open-ended responses about participants' thoughts during Survey 2. Out of 218 total responses (across 81 scenarios), 118 of them (across 63 scenarios) described motivations other than the provided ones.

Participants provided more comments explaining the negative driving forces behind their decisions—with the majority (112 out of 118 comments) explaining why they chose not to help. These additional motivations were categorized into eight distinct groups (Supplementary Table 4). Among the most frequently mentioned were Contextual Considerations (i.e., help depends on the specific feature of the helpee and event), Help Could Lead to Bad Consequences (i.e., the outcome of help may not be ideal), Capacity (i.e., one may not be capable of help), and Prefer Other Options (i.e., prefer alternative actions to help over the provided action). On the other hand, motivations such as Similarity (i.e., similar personal experience creates connections with the helpee) had a positive impact on help decisions. Nevertheless, Contextual Considerations and Prefer Other Options could potentially lead to help decisions if circumstances were to change.

## Discussion

In this study, we characterized people's willingness to help across a broad variety of everyday scenarios and provided a computational account of the psychological processes underlying helping decisions. We found that people decide whether to help or not mainly based on two orthogonal

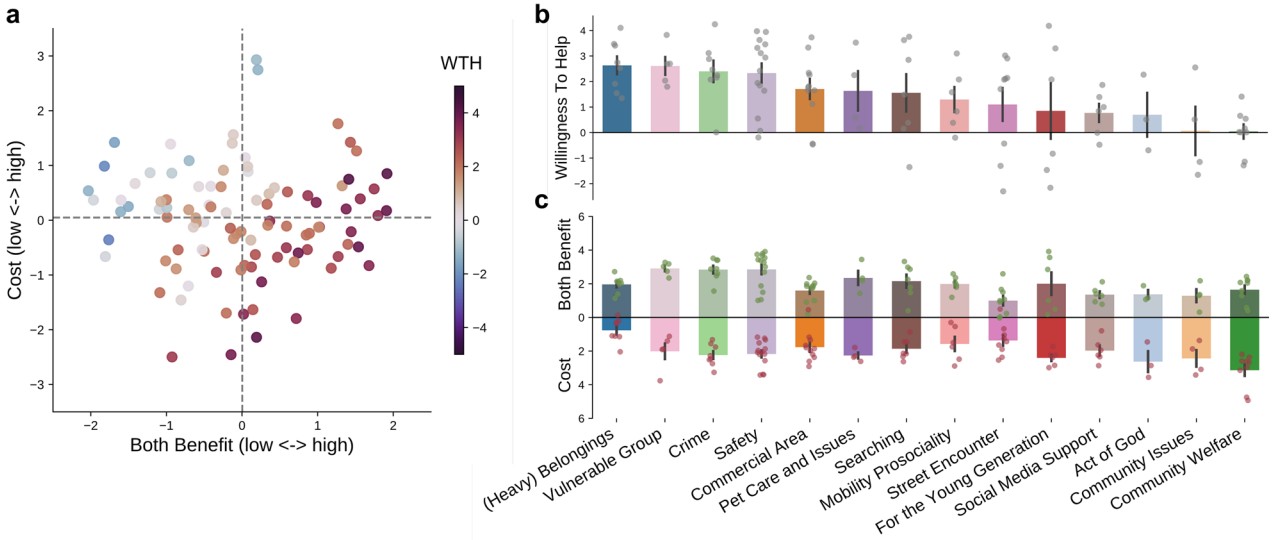

**Fig. 6 | Associations among three latent spaces. a** Decision space vs. motivation space. Scenarios plotted in the motivation space were color-coded by the average WTH. Red indicates positive WTH, blue indicates negative WTH. **b** Decision space vs. semantic space. Bar plots show the WTH of each semantic category. Error bars denote the standard error. Each dot represents a scenario. **c** Motivation space vs. semantic space. Bar plots show the Benefit (upper panel) and Cost (lower panel) of each semantic category. Error bars denote the standard error. Each green dot represents the Benefit of a scenario, and each red dot represents the Cost of a scenario. Note that for better visualization, we transformed the factor scores to non-negative, thus the size of the bar represents the absolute magnitude of Benefit and Cost. Number of scenarios in each semantic category: belongings ($n=8$), vulnerable group ($n=5$), crime ($n=8$), safety ($n=14$), commercial area ($n=11$), pet care and issues ($n=4$), searching ($n=7$), mobility prosociality ($n=6$), street encounter ($n=9$), for the young generation ($n=6$), social media support ($n=6$), act of God ($n=3$), community issues ($n=4$), community welfare ($n=9$).

factors: the benefit of help for both the people in need and themselves, as well as their own cost of help. Despite such a universal decision algorithm, individual computations varied: individuals with higher punishment sensitivity placed greater emphasis on the cost (relative to benefit), while those higher in agreeableness or empathy showed a stronger general disposition to help regardless of the cost/benefit tradeoff. In addition, a thorough characterization of scenarios demonstrated how different semantics of the scenarios are associated with various motivations, as well as the helping decisions.

### Modeling helping decisions in ecologically valid scenarios

Over the past decade, psychologists and economists started applying decision-making theories to explain social decisions, including helping[52,53]. The subjective utility of helping is often modeled as a weighted combination of cost and benefit. Our analysis of the motivation factors confirmed that these two variables indeed uniquely contribute to the helping decisions. Among the tested models, the best-performing one describes the subjective utility of help as a linear weighted sum of cost and benefit in addition to a baseline helping propensity. Crucially, the use of a single weight parameter for cost-benefit tradeoffs outperformed the model with independent weights, indicating the inter-dependency of one's focus on cost and benefit—those who care more about cost also care less about benefit.

That said, one needs to be cautious when interpreting such weight - since the cost and benefit ratings were derived from the group consensus, we cannot dissociate two possibilities: a high $w_{cost}$ may suggest a stronger weighting of the perceived cost (while the subjective estimation of cost is the same as the consensus) or an elevated perception of the cost relative to the consensus. Our finding was in part consistent with Hu et al.[11], who also found that a single-weight cost-benefit model was better than a double weights model; yet they reported a nonlinear transformation of benefit which did not emerge in our data. Both of our findings indicate a zero-sum tradeoff between costs and benefits: when costs become more prominent, benefits are less influential (and vice versa). This has practical implications for prosocial interventions: when promoting charitable giving,

volunteerism, and emergency assistance, reducing the perceived effort or risks can be as crucial as emphasizing positive outcomes.

Our study offers a mathematical quantification of helping decisions across a diverse set of real-world scenarios, extending beyond the well-controlled laboratory paradigms typically used in prior research, where participants repeatedly make the same type of decisions (e.g., giving money) in response to pre-defined levels of motivational variables[8,9,11,38]. By incorporating greater flexibility in presenting situational variables, we enhance the ecological validity of our findings. Importantly, our cost and benefit dimensions are grounded in participants' naturalistic priors (i.e., knowledge learned through real-life experience but not the experiment)[16,17], still, we observed strong convergence in these judgments and their influence on decisions—participants reached high consensus in motivation ratings, and the cost-benefit model showed good predictive performance to the helping behavior. As participants made helping decisions before being introduced to our hypothesized dimensions, these patterns cannot be explained by social desirability bias. Together, these results provide empirical evidence that the cost-benefit tradeoff principle operates in naturalistic decision-making, not only in well-defined, unambiguous laboratory settings.

### Comprehensive characterization of everyday helping situations

"Helping" encompasses a wide array of behaviors. There have been various taxonomies describing helping, yet some definitions can be arbitrary and lack empirical support[54-56]. To better understand the relationships among different helping situations without pre-assumptions, we adopted a purely data-driven approach. An early attempt of this approach used similarity ratings (between scenarios) to uncover three latent dimensions—planned vs. spontaneous, serious vs. not serious, indirect vs. direct intervention[57]. Here, we decomposed the similarity into three perspectives: willingness to help in the scenario, strengths of possible motivations, and semantic content of the scenario narratives, and represented the scenarios in the corresponding spaces. In the decision space, scenarios were distributed along a continuum of average WTH levels; in the motivation space, scenarios varied at two orthogonal factors (cost and benefit) and are thus classified into four quadrants (high vs. low cost * high vs. low benefit); in the semantic space,

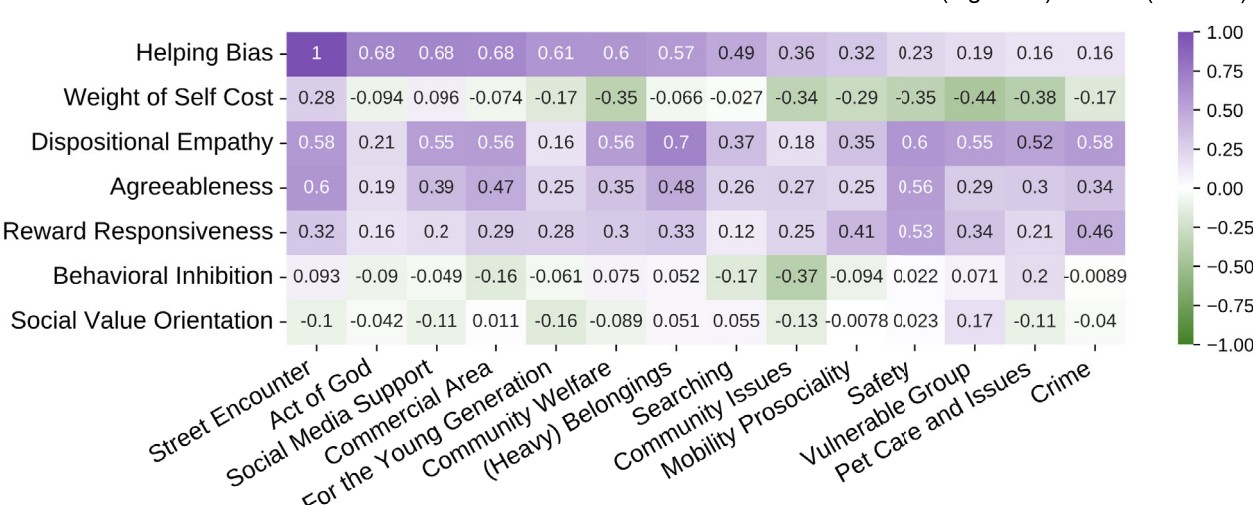

**Fig. 7 | Context-dependent effect of personal traits on helping behavior.** Each cell represents the Cohen's d (across N = 215 participants) for the WTH difference between individuals high in a certain trait and individuals low in a certain trait (by rows), for a particular semantic category of scenarios (by columns). Positive d (red) indicates positive associations between a trait and the WTH, negative d (green) indicates negative associations between a trait and the WTH.

scenarios that share similar objectives, actions, or environmental settings were clustered together. Leveraging the cross-scenario similarities derived from the three spaces, we further dissociated the unique contributions of the motivation space and semantic space in driving helping behaviors: across individuals, consensus on the costs and benefits was related to consensus on the willingness to help, whereas within individuals, semantically similar situations were evaluated more consistently.

Although the spaces were established based on non-overlapping variables, we observed inherent relationships: different semantic categories feature different levels of cost and benefit, which potentially contribute to different WTH ratings. For example, intervening in a crime scenario may involve high personal risk (high cost) but offer significantly rewarding outcomes (e.g., saving someone's life, boosting one's reputation)—resulting in a generally high willingness to help.

While constructing the motivation space, we initially proposed 12 different dimensions based on existing theories. Although all the dimensions were highly relevant to the two identified factors and thus predictive of WTH, the amount of unique latent dimensions was smaller than expected. For example, we did not dissociate the benefit for self and other, since scenarios that were evaluated to bring big benefits/avoid big harm to the helpee(s) were also rated high on the self-reputation improvement, mood enhancement, reciprocity, etc. One possibility may lie in the design: participants rated all the motivation dimensions simultaneously, so the evaluation of one dimension may affect ratings of other dimensions, or share correlated error. Future studies may optimize the design and only allow one dimension to be rated by one participant. Still, the observed correlations do not imply that every participant perceived all motivations as equivalent: depending on the participant, some of the motivations can be clearly distinguished from others.

**Distinct components of helping map to distinct personality traits**
Many studies have investigated how personalities influence helping behavior, and identified a set of key relevant personality dimensions such as dispositional empathy[29,58], social value orientation[30,49], interpersonal trust[32,59], and agreeableness[31,60]. Most studies represented the helping behavior using a single variable (e.g., willingness to help, percentage of helping behaviors, etc.), neglecting the multi-step nature of decision-making. Leveraging computational models, we not only replicated the association between the overall WTH and key personality traits, but also isolated sub-components of the decision processes: baseline helping

propensity was predicted by dispositional empathy and agreeableness; whereas punishment sensitivity, despite no direct correlation with the overall WTH, influenced decisions indirectly by modulating the weight of cost (i.e., people who are more sensitive to punishment or negative consequences are more hesitant to help when the cost is high).

Interestingly, we found that the model-based helping-specific traits and other questionnaire-assessed personality traits showed context-dependency when predicting helping in different contexts. While some of the traits acted as a domain-general predictor of WTH across the majority types of helping scenarios (e.g., helping bias, dispositional empathy, agreeableness), others specifically affected smaller subsets of contexts. For example, for situations that are typically high benefit and high cost (e.g., helping the vulnerable group), individuals with different levels of helping bias did not show much different decisions; whereas individuals with higher (vs. lower) weights of self-cost tended to provide less help. These findings further demonstrated that helping is not a uni-dimensional construct - instead, the contextual features of a helping scenario will elicit varied responses along multiple trait dimensions.

One thing worth noting is that we did not find any statistically significant association between WTH and social value orientation (SVO), even after breaking down the types of scenarios. As SVO directly measures how one allocates resources between self and other, previous studies that demonstrated the associations all defined 'helping' using paradigms that involve a redistribution of the same type of resource (i.e., money[30,49], and time[61]). In our paradigm, helping is conceptualized as sacrificing some form of one's own resource to increase (or reduce the use of) other's resource (potentially) in another form. Apart from money and time, these resources have more diverse forms, such as physical strength, effort, and knowledge that are not directly comparable even within the same category (e.g., to lift up the same bag, a young kid needs more effort than an adult). Therefore, it may not involve the same resource allocation process as measured in SVO, and that SVO may be a better predictor of the prosocial behaviors observed from social dilemmas involving the distribution of shared resources[62,63] rather than everyday helping.

**Limitation**
Despite the strengths of our approach, the current study has several limitations. First, while modeling the helping decisions, we utilized the benefit and cost scores derived from the group-level consensus ratings. Since different people may have different estimates of the cost/benefit, the model

fitting could be improved if the individual ratings were applied. However, collecting full motivation ratings for each participant across all 100 scenarios was not feasible, especially in a remote setting where data quality must be maintained. Future studies could streamline the design by using only the two latent dimensions and collecting personalized ratings across all scenarios.

Second, despite the use of a large, diverse scenario set, our measured willingness to help is still hypothetical. Real-world helping scenarios are much more complicated, with many social and environmental features interacting with each other. For example, the existence of other individuals could possibly lead to one's reduced likelihood of helping—a phenomenon called 'bystander effect'[48]; the identity of the helpee—whether they come from a similar socioeconomic background as the individual, would also make a difference[9,32]. As such, there may be discrepancies between participants' responses in online surveys and actual behaviors. Future work should aim to incorporate richer contextual details in scenario design and validate these models using field data collected in the real world.

## Data availability
All the de-identified data are publicly available at https://github.com/wuqy052/Help_Decision_Naturalistic and https://doi.org/10.17605/OSF.IO/EN3T9.

## Code availability
All the analysis code is publicly available at https://github.com/wuqy052/Help_Decision_Naturalistic and https://doi.org/10.17605/OSF.IO/EN3T9.

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

## Acknowledgements

This study was funded by Honda Research Institute, USA. We thank Mungyeong Choe and Yuchen Yan for constructive feedback during the study design, and thank Yue Xu for helpful discussions on the methods.

## Author contributions

Q.W. conceptualized the study, led data curation and formal analysis, wrote the original draft, and revised the manuscript. M.S. conceptualized the study, contributed to formal analysis, review and editing, and supervised the project. J.A. contributed to study conceptualization, review and editing, and supervised the project. D.D. contributed to data collection and curation, review and editing, and supervised the project. D.T.conceptualized the study, reviewed and edited the draft, and supervised and administered the project. E.M.-P. contributed to funding acquisition, project supervision, and manuscript review and editing.

## Competing interests

This study was funded by Honda Research Institute, USA (HRI). E.M.-P. is an employee of HRI, M.S., D.T., J.A. were employees of HRI, and Q.W. was an intern of HRI at the time of the work. D.D. is a project collaborator at the University of Michigan. HRI determined the broader project direction, but did not participate in study design, data collection and analysis, decision to publish or preparation of the manuscript.
