## [Transparent Peer Review file · Communications Psychology]

Prosocial decisions in naturalistic helping scenarios are predicted by cost-benefit tradeoffs and individual disposition

Corresponding Author: Ms Qianying Wu

Version 0:

Decision Letter:

Dear Ms Wu,

Thank you for your patience during the peer-review process. Your manuscript titled "Why we help strangers: decoding everyday prosocial decisions through a cost-benefit modeling" has now been seen by 2 reviewers, and I include their comments at the end of this message. They find your work of interest but raised some important points. We are interested in the possibility of publishing your study in Communications Psychology, but would like to consider your responses to these concerns and assess a revised manuscript before we make a final decision on publication.

We therefore invite you to revise and resubmit your manuscript, along with a point-by-point response to the reviewers. Please highlight all changes in the manuscript text file.

Editorially, we consider it important that the reviewers' methodological and presentational concerns (in particular about conceptual clarity) are fully addressed. Please also ensure that the paper makes it clear how the present study confirms, contradicts, or extends the existing literature. Novelty claims must be avoided, and the presentation should remain matter-of-fact.

I am attaching an Editorial Requests Table that details critical reporting requirements for the revised manuscript. Please attend to each item and ensure your manuscript is fully compliant. If your revised manuscript is not aligned with these requests on major issues, such as those concerning statistics, it may be returned to you for further revisions without re-review.

Please submit the following items:

- Revised manuscript
- Point-by-point response to the referees' comments
- Cover letter (as a separate document)
- <https://www.nature.com/documents/nr-reporting-summary.pdf> Nature Research Reporting Summary
- Completed Editorial Request Table (attached).

via this link: Link Redacted .

Additional guidance is available in our style and formatting guide Communications Psychology formatting guide.

Best regards,

Marika Schiffer

Marika Schiffer, PhD
Chief Editor
Communications Psychology

REVIEWER EXPERTISE:

Reviewer #1 & Reviewer #2: prosocial behaviour; computational modeling; RSA

REVIEWER REPORTS:

Reviewer #1 (Remarks to the Author):

This paper presents several naturalistic helping scenarios and develops a cost-benefit model to analyze human prosocial behavior. The research is both interesting and valuable. Below are some suggestions for improvement:

Introduction

1. The authors discuss various psychological drivers of prosocial behavior in the Introduction; however, their experimental design and data analysis appear to focus primarily on the cost-benefit tradeoff, rather than these multiple drivers. This disconnect weakens the logical coherence of the paper. I suggest the authors revise the Introduction to better align with the core focus of the study and clearly state the main objectives and framework of the present work.
2. I also recommend reorganizing the Introduction to emphasize the innovations and significance of this study in comparison to previous research. Highlighting what is novel and why it matters would help situate the study more effectively within the existing literature.

Methods and Results

3. The authors conducted a detailed classification of helping scenarios, which is commendable. However, this part seems somewhat tangential to the main focus of the study. Moreover, these findings raise an interesting question about whether individual differences in personality might influence helping behavior across different types of scenarios. The authors could consider exploring this possibility with further analyses.
4. Do different psychological drivers—such as empathy, reciprocity, and social norms—carry different weights in predicting helping behavior? It would be valuable for the authors to examine and report on the relative contributions of these factors.

Reviewer #2 (Remarks to the Author):

This study contributes to understanding altruistic motivation by examining everyday helping scenarios. The authors conducted two online studies ($N_1=215$; $N_2=140$) to assess helping motivations, using EFA to identify two latent factors ("benefit to both helper/helped" and "cost to the helper") and modeling decisions as a linear weighted sum of these factors alongside dispositional bias. The correlations between model parameters and empathy/agreeableness provide individual-difference perspectives. While the shift from economic games to real-life contexts is commendable, and writing is clear, I have the following comments

1. While the focus on daily-life scenarios effectively extends traditional economic paradigms, the core computational framework (weighing costs/benefits) aligns closely with established models (e.g., Hu et al., 2021, *JNeurosci*). To highlight the study's unique contribution, the Introduction and Discussion could more explicitly articulate how naturalistic contexts

reveal new mechanisms or boundary conditions not captured in lab-based tasks. Clarifying how real-world complexity advances theory beyond prior work would strengthen the manuscript's impact.

2. Page 32: The rationale for binarizing Likert-scale ratings (collapsing into "help" vs. "no help") is noted. However, treating willingness-to-help as a continuous measure might better capture nuanced decision thresholds. Could the authors also model the raw ratings to assess robustness? A comparison of results using both approaches would strengthen methodological transparency.

3. Page 31: The choice of varimax rotation assumes orthogonal (uncorrelated) factors. Given that motivations like "cost" and "benefit" may theoretically interrelate, could the authors consider promax rotation (allowing correlated factors) and report whether results meaningfully differ? This would ensure the factor structure optimally reflects potential motivational interdependencies.

4. The analyses linking decision, motivation, and semantic spaces are technically sound. To enhance conceptual clarity, the manuscript might more explicitly state how these associations provide novel insights beyond confirming expected relationships between each latent spaces.

Version 1:

Decision Letter:

Dear Ms Wu,

Your manuscript titled "Why we help strangers: decoding everyday prosocial decisions through cost-benefit modeling" has now been seen by our reviewers, whose comments appear below. In light of their advice I am delighted to say that we are happy, in principle, to publish a suitably revised version in Communications Psychology.

We therefore invite you to revise your paper one last time to address the remaining concerns of our reviewers and a list of editorial requests. At the same time we ask that you edit your manuscript to comply with our format requirements and to maximise the accessibility and therefore the impact of your work.

EDITORIAL REQUESTS:

SUBMISSION INFORMATION:

OPEN ACCESS:

* DATA AVAILABILITY:

Link Redacted

Best regards,

Jennifer Bellingtier

Jennifer Bellingtier, PhD
Senior Editor
Communications Psychology

REVIEWERS' EXPERTISE:

Reviewer #2: prosocial behaviour; computational modeling; RSA

REVIEWERS' COMMENTS:

Reviewer #2 (Remarks to the Author):

The authors have addressed all my comments, and I recommend this manuscript for publication.

** Visit Nature Research's author and referees' website at a

href="http://www.nature.com/authors">www.nature.com/authors for information about policies, services and author benefits**

We appreciate the thoughtful and constructive comments of both reviewers, which we believe have led to significant improvements in the manuscript. Below please find our detailed responses to their concerns.

Reviewer #1 (Remarks to the Author):

This paper presents several naturalistic helping scenarios and develops a cost-benefit model to analyze human prosocial behavior. The research is both interesting and valuable. Below are some suggestions for improvement:

Introduction

1. The authors discuss various psychological drivers of prosocial behavior in the Introduction; however, their experimental design and data analysis appear to focus primarily on the cost-benefit tradeoff, rather than these multiple drivers. This disconnect weakens the logical coherence of the paper. I suggest the authors revise the Introduction to better align with the core focus of the study and clearly state the main objectives and framework of the present work.

Thank you for your suggestions. Based on this suggestion and comment #2, as well as Reviewer 2's similar suggestions in their comment #1, we have revised the Introduction thoroughly to highlight the main objectives of the study and our contributions to the research field. We now emphasize that we aim at generalizing the well-established cost-benefit tradeoff framework to model helping decisions in more ecologically valid decisions, in two key aspects: (1) use real life scenarios to reduce artificiality and increase naturality in the decision context; and (2) measure subjectively perceived cost/benefits through motivations that are social and affective in nature, to better account for the complexity of the social decisions.

Also, regarding why we introduce multiple psychological drivers, we clarified that these psychological drivers may be domain-specific motivations that people use in social decisions, rather than domain-general motivations such as monetary cost and reward. Therefore, we operationalized these social and affective motivations as a set of subjective rating questions. How we selected these motivational dimensions are based on existing psychological theories that we briefly introduced and thus justified to the readers.

Below are the revised paragraphs:

“(page 4) Moving from a simplicity-focused approach to embracing real-world complexity, we extended the definition of costs and benefits beyond domain-general valuation currencies such as money and effort. As helping decisions largely involve social cognition and perception, we proposed that domain-specific

motivations that are social and affective in nature play important roles in driving these decisions. These motivations reflect three broad theoretical traditions in the psychology of prosocial behavior. First, from an evolutionary perspective, helping unrelated individuals may promote long-term benefits through reciprocal altruism or competitive altruism, by fostering future cooperation or enhancing one's reputation and social status (Grafen, 1990; Penner. et al., 2005; Roberts, 1998; Trivers, 1971; Wedekind & Braithwaite, 2002). Second, emotional theories such as the 'negative state relief model' emphasize internal affective benefits: helping alleviates personal distress caused by witnessing others suffer, mitigates guilt, and improves mood (Batson et al., 1989; Cialdini & Kenrick, 1976; Fan et al., 2011). Third, socialization perspectives highlight the role of moral norms and social expectations, suggesting that people help others to maintain a positive self-concept, fulfill internalized social responsibilities, and avoid social punishments (e.g., self-concept threats, reputation damage) (Schaller & Cialdini, 1988; Schwartz, 1977; Schwartz & Howard, 1984; Young et al., 2012). Although these motivations have each been examined in isolation, they are rarely considered together within a single computational framework.

Here, we hypothesized that helping decisions in everyday contexts still follow a cost-benefit tradeoff model but involve an expanded motivational architecture. We operationalized this using 12 motivational dimensions (guilt relief, self-concept maintenance, expected reciprocity, etc.), rated by a group of online participants. Costs and benefits associated with the scenarios were constructed through exploratory factor analysis and subsequently applied to a decision model to predict helping decisions collected from a broader sample.”

2. I also recommend reorganizing the Introduction to emphasize the innovations and significance of this study in comparison to previous research. Highlighting what is novel and why it matters would help situate the study more effectively within the existing literature.

As mentioned in the previous comment, we have incorporated the reviewers' suggestions and highlighted the innovations and significance of the current research.

(page 3, Introduction) “Traditional economic theories frame decision-making as a cost-benefit tradeoff, assuming that individuals prioritize self-interest by maximizing personal benefit and minimizing cost (Drèze & Stern, 1987; Von Neumann & Morgenstern, 1944). Later research extended this framework by showing that individuals also consider others' welfare in decisions: the subjective utility of helping has been successfully modeled as an integration of self- and

other-related outcomes (Fehr & Krajbich, 2014; Hayashi & Tahmasbi, 2020; Hu et al., 2021; Hutcherson et al., 2015; Liu et al., 2020; Lockwood et al., 2022; Morelli et al., 2015; Q. Wu & O'Doherty, 2025). For instance, one study modeled decisions to forgo monetary rewards (cost) to prevent others from hearing aversive noise (benefit) (Hu et al., 2021), while another examined effort exertion (cost) to earn money for others (benefit) (Lockwood et al., 2022). While these studies provided rigorous support for cost-benefit models, they relied on artificial, repetitive helping scenarios and objectively controlled costs and benefits. Such simplifications, although necessary for internal validity, limit our understanding of how cost-benefit tradeoffs operate in everyday settings.

To address this gap, we developed a novel task grounded in naturalistic helping scenarios. Our aim was two-fold: (1) to test whether cost-benefit computations generalize to real-world situations, and (2) to identify domain-specific motivational mechanisms that emerge in prosocial contexts. Our design enhances ecological validity (Holleman et al., 2020) along two key axes: from artificiality to naturality, and from simplicity to complexity.”

(page 4, Introduction) “Moving from artificiality toward naturality, we emphasize that real-world helping often relies on rich contextual knowledge rather than explicit numerical information. Most laboratory paradigms intentionally minimize such priors by using abstract, well-defined tradeoffs (e.g., exact amounts of money or effort) (Holleman et al., 2020; Wise et al., 2024). In contrast, daily helping requires drawing on common-sense understanding of human behavior and social situations (Markman, 2018). To approximate this, we created a stimulus set of 100 brief narratives depicting diverse helping scenarios, primarily involving strangers. Instead of assigning fixed monetary or physical costs, we collected subjective ratings across a wide array of motivational dimensions to capture intuitive judgments in natural settings.”

(page 23, Discussion) “Our study offers a mathematical quantification of helping decisions across a diverse set of real-world scenarios, extending beyond the well-controlled laboratory paradigms typically used in prior research, where participants repeatedly make the same type of decisions (e.g., giving money) in response to pre-defined levels of motivational variables (Hayashi & Tahmasbi, 2020; Hu et al., 2021; Hutcherson et al., 2015; Lynch & Cohen, 1978). By incorporating greater flexibility in presenting situational variables, we enhance the ecological validity of our findings. Importantly, our cost and benefit dimensions are grounded in participants’ naturalistic priors (i.e., knowledge learned through real life experience but not the experiment) (Markman, 2018; Wise et al., 2024), still, we observed strong convergence in these judgments and their influence on decisions - participants reached high consensus in motivation ratings, and that

the cost-benefit model showed good predictive performance to the helping behaviors. As participants made helping decisions before being introduced to our hypothesized dimensions, these patterns cannot be explained by social desirability bias. Together, these results provide empirical evidence that the cost-benefit tradeoff principle operates in naturalistic decision-making, not only in well-defined, unambiguous laboratory settings.”

The rest are quoted in the comment #1 response.

Methods and Results

3. The authors conducted a detailed classification of helping scenarios, which is commendable. However, this part seems somewhat tangential to the main focus of the study. Moreover, these findings raise an interesting question about whether individual differences in personality might influence helping behavior across different types of scenarios. The authors could consider exploring this possibility with further analyses.

We agree that a more thorough investigation of how the personalities might modulate willingness to help (WTH) is valuable. Therefore, we performed an exploratory analysis comparing how each trait (5 questionnaire-assessed traits + 2 model parameter) affected WTH across different semantic categories. We calculated effect sizes (Cohen's d) of how WTH ratings differ between individuals high in a particular trait (above median) vs. individuals low in a particular trait (below median), across all the traits, and all the semantic categories. A higher Cohen's d indicates a stronger impact of the trait to the given scenario category.

The result provides 3 insights: (1) we observed an overall stronger effect of helping bias, dispositional empathy, agreeableness, and reward responsiveness in impacting all kinds of scenarios, consistent with our previous findings that these traits are correlated with the overall WTH. (2) There exist variabilities across scenario categories, such that one trait can selectively affect a subset of categories more than the rest. (3) there also exist variabilities across traits, such that WTH of the same semantic category may be better predicted by some traits than others. Please find below our detailed description on these results.

(page 42, Methods) “In addition, we explored how different traits (questionnaire-assessed and model-derived) influence helping behaviors across different categories of scenarios. For each trait, we divided the participants into a high trait group and a low trait group through a median split. We then quantified the WTH differences of each semantic category of scenarios between high trait and low trait groups through Cohen's d . According to existing criteria (Cohen, 2013), a d (absolute value) of 0.2 is considered small, 0.5 is medium, and 0.8 is large. A

positive d means individuals higher in a trait are more likely to help; and a negative d means individuals lower in a trait are more likely to help.”

(page 19-21, Results) **“Personal traits differentially modulate helping in different types of scenarios**

As we generated meaningful semantic interpretations of the helping scenarios, we further asked whether personal traits influence helping behaviors in a context-dependent manner - that is, whether some of the traits selectively impact WTH in some types of the helping situations but not others. We investigated this nuanced relationship by comparing the differences in WTH ratings (of each semantic category) between individuals high in and low in each trait (including five questionnaire-assessed personality traits and two model-based parameters, Figure 7). Consistent with our previous analyses, the helping bias, dispositional empathy, agreeableness and reward responsiveness showed more than small effect sizes impacting most scenario categories, confirming their role as a general indicator of helping intentions. Beyond such general effect of traits, we also observed context-dependency between traits and scenarios: a given trait could have bigger effects on a subset of scenarios than others; and WTH of a given scenario category could be better explained by a few traits.

More specifically, the two model parameters showed complementary effects among scenario types: while the helping bias showed greater effects in predicting most types of scenarios (Cohen's d range: 0.16 ~ 1), the weight of self cost is more predictive of helping in the 'vulnerable group' (Cohen's $d = -0.44$), 'pet care and issues' (Cohen's $d = -0.38$) and 'safety' (Cohen's $d = -0.35$) scenarios - scenarios that are typically high benefit and high cost. Similarly, across the five personality traits, dispositional empathy showed greater effects in most scenarios (Cohen's d range: 0.18 ~ 0.70), yet behavioral inhibition had a stronger association with 'community issues' (Cohen's $d = -0.37$) scenarios, which feature high cost and low benefit. Social value orientation, on the other hand, showed minimal effects regardless of the context (Cohen's d range: -0.16 ~ 0.17).”

(page 25, Discussion) “Interestingly, we found that the model-based helping-specific traits and other questionnaire-assessed personality traits showed context-dependency when predicting helping in different contexts. While some of the traits act as a domain-general predictor of WTH across the majority types of helping scenarios (e.g., helping bias, dispositional empathy, agreeableness), others specifically affect smaller subsets of contexts. For example, in situations that are typically high benefit and high cost (e.g., helping the vulnerable group), individuals who have different levels of helping bias do not show much different decisions; whereas individuals with higher (vs. lower) weights of self cost tend to

provide less help. These findings further demonstrate that helping is not a uni-dimensional construct - instead, the contextual features of a helping scenario will elicit varied responses along multiple trait dimensions.”

Figure 7. Context-dependent effect of personal traits on helping behaviors. Each cell represents the Cohen’s d for the WTH difference between individuals high in a certain trait (by rows) and individuals low in a certain trait, for a particular semantic category of scenarios (by columns). Positive d indicates positive associations between a trait and the WTH, negative d indicates negative associations between a trait and the WTH.

4. Do different psychological drivers—such as empathy, reciprocity, and social norms—carry different weights in predicting helping behavior? It would be valuable for the authors to examine and report on the relative contributions of these factors.

Thank you for suggesting this analysis. While it would be ideal to assess the relative contribution of each psychological driver in a unified model, in practice such modeling analysis is not feasible with the current data, due to the following reasons: (1) all the drivers are highly correlated, as reported in Supplementary Figure 3a, thus including all of them in the same model will have a severe multicollinearity issue; (2) each participant had 50 decision trials, thus the sample size is too small to fit a model with at least 12 parameters (each weight for a driver, the rule-of-thumb is usually $N_{\text{sample}} = 10 * n_{\text{trials}}$).

Supplementary Figure 3. (a) Correlations among the ratings of 12 original motivation dimensions. Orange indicates positive loadings, and blue indicates negative loadings. The color intensity is proportional to the loading strength.

However, to potentially answer this question, we conducted a new analysis, in which we assessed the contribution of each individual motivation, respectively (instead of including all in a single model). We ran linear mixed-effect models predicting WTH ratings with each motivational dimension as a fixed effect, and participant as a random effect (intercept + slope). We then compared their explained variance as defined marginal R^2 (for fixed effect of motivation) and conditional R^2 (for both fixed and random effects). We found that all the motivations explained moderate to high levels of variance of WTH ratings. Among them peers think one should help (injunctive norm) explained most variance (marginal $R^2 = 0.358$, conditional $R^2 = 0.817$), and other's bad consequence explained the least variance (marginal $R^2 = 0.214$, conditional $R^2 = 0.490$). We have included this new information in the manuscript.

(page 35, Methods) "**Associations between motivational dimensions and WTH**

We assessed how well each motivation predicts the WTH across scenarios through linear mixed-effects models. As the motivational dimensions highly correlate with each other, to avoid the problem of multi-collinearity, we ran separate models for each motivation in the formula of $WTH = 1 + \text{Motivation} + (1 + \text{Motivation} | \text{sub})$. The model included a motivation as a fixed effect and allowed for random intercepts and random slopes for each participant.

We quantified the model fits using marginal R^2 (variance attributed to fixed effects only) and conditional R^2 (variance attributed to both fixed effects and random effects) (Nakagawa & Schielzeth, 2013).”

(page 8, Results) “We assessed the predictive performance of each motivational feature to the WTH ratings, and found that all hypothesized motivation dimensions explained a medium to large portion of WTH variances across scenarios and individuals. Despite an overall good performance, variability exists: among all motivations, injunctive norm (peers think one should help) explained most variance (marginal $R^2 = 0.358$, conditional $R^2 = 0.817$, derived from a linear mixed effect model), and other’s cost if not help explained the least variance (marginal $R^2 = 0.214$, conditional $R^2 = 0.490$, Supplementary Figure 2).”

Supplementary Figure 2. WTH variance explained by each motivational dimension. Explained variance was assessed using marginal R^2 (for fixed effect of motivation) and conditional R^2 (for both fixed effect and random effect). The bars are superimposed, thus the magnitude of each R^2 is shown in its corresponding x axis values.

Reviewer #2 (Remarks to the Author):

This study contributes to understanding altruistic motivation by examining everyday helping scenarios. The authors conducted two online studies ($N_1=215$; $N_2=140$) to assess helping motivations, using EFA to identify two latent factors ("benefit to both helper/helped" and "cost to the helper") and modeling decisions as a linear weighted sum of these factors alongside dispositional bias. The

correlations between model parameters and empathy/agreeableness provide individual-difference perspectives. While the shift from economic games to real-life contexts is commendable, and writing is clear, I have the following comments

1. While the focus on daily-life scenarios effectively extends traditional economic paradigms, the core computational framework (weighing costs/benefits) aligns closely with established models (e.g., Hu et al., 2021, JNeurosci). To highlight the study's unique contribution, the Introduction and Discussion could more explicitly articulate how naturalistic contexts reveal new mechanisms or boundary conditions not captured in lab-based tasks. Clarifying how real-world complexity advances theory beyond prior work would strengthen the manuscript's impact.

Thank you for your helpful feedback. As stated above, we have incorporated your suggestion as well as Reviewer 1's suggestions in the Introduction and Discussion to highlight why our research on daily-life scenarios are valuable in complementing existing lab-based tasks.

(page 3-5, Introduction) "While these studies provided rigorous support for cost-benefit models, they relied on restricted, repetitive helping scenarios and objectively controlled costs and benefits. Such simplifications, although necessary for internal validity, limit our understanding of how cost-benefit tradeoffs operate in everyday settings. To address this gap, we developed a novel task grounded in naturalistic helping scenarios. Our aim was two-fold: (1) to test whether cost-benefit computations generalize to real-world situations, and (2) to identify domain-specific motivational mechanisms that emerge in prosocial contexts. Our design enhances ecological validity (Holleman et al., 2020) along two key axes: from artificiality to naturality, and from simplicity to complexity.

Moving from artificiality toward naturality, we emphasize that real-world helping often relies on rich contextual knowledge rather than explicit numerical information. Most laboratory paradigms intentionally minimize such priors by using abstract, well-defined tradeoffs (e.g., exact amounts of money or effort) (Holleman et al., 2020; Wise et al., 2024). In contrast, daily helping requires drawing on common-sense understanding of human behavior and social situations (Markman, 2018). To approximate this, we created a stimulus set of 100 brief narratives depicting diverse helping scenarios, primarily involving strangers. Instead of assigning fixed monetary or physical costs, we collected subjective ratings across a wide array of motivational dimensions to capture intuitive judgments in natural settings.

Moving from a simplicity-focused approach to embracing real-world complexity, we extended the definition of costs and benefits beyond domain-general valuation currencies such as money and effort. As helping decisions largely involve social cognition and perception, we proposed that domain-specific motivations that are social and affective in nature play important roles in driving these decisions. These motivations reflect three broad theoretical traditions in the psychology of prosocial behavior. First, from an evolutionary perspective, helping unrelated individuals may promote long-term benefits through reciprocal altruism or competitive altruism, by fostering future cooperation or enhancing one's reputation and social status (Grafen, 1990; Penner. et al., 2005; Roberts, 1998; Trivers, 1971; Wedekind & Braithwaite, 2002). Second, emotional theories such as the 'negative state relief model' emphasize internal affective benefits: helping alleviates personal distress caused by witnessing others suffer, mitigates guilt, and improves mood (Batson et al., 1989; Cialdini & Kenrick, 1976; Fan et al., 2011). Third, socialization perspectives highlight the role of moral norms and social expectations, suggesting that people help others to maintain a positive self-concept, fulfill internalized social responsibilities, and avoid social punishments (e.g., self-concept threats, reputation damage) (Schaller & Cialdini, 1988; Schwartz, 1977; Schwartz & Howard, 1984; Young et al., 2012). Although these motivations have each been examined in isolation, they are rarely considered together within a single computational framework.

Here, we hypothesized that helping decisions in everyday contexts still follow a cost-benefit tradeoff model but involve an expanded motivational architecture. We operationalized this using 12 motivational dimensions (guilt relief, self-concept maintenance, expected reciprocity, etc.), rated by a group of online participants. Costs and benefits associated with the scenarios were constructed through exploratory factor analysis and subsequently applied to a decision model to predict helping decisions collected from a broader sample.”

(page 23, Discussion)” Our study offers a mathematical quantification of helping decisions across a diverse set of real-world scenarios, extending beyond the well-controlled laboratory paradigms typically used in prior research, where participants repeatedly make the same type of decisions (e.g., giving money) in response to pre-defined levels of motivational variables (Hayashi & Tahmasbi, 2020; Hu et al., 2021; Hutcherson et al., 2015; Lynch & Cohen, 1978). By incorporating greater flexibility in presenting situational variables, we enhance the ecological validity of our findings. Importantly, our cost and benefit dimensions are grounded in participants' naturalistic priors (i.e., knowledge learned through real life experience but not the experiment) (Markman, 2018; Wise et al., 2024), still, we observed strong convergence in these judgments and their influence on

decisions - participants reached high consensus in motivation ratings, and that the cost-benefit model showed good predictive performance to the helping behaviors. As participants made helping decisions before being introduced to our hypothesized dimensions, these patterns cannot be explained by social desirability bias. Together, these results provide empirical evidence that the cost-benefit tradeoff principle operates in naturalistic decision-making, not only in well-defined, unambiguous laboratory settings.”

2. Page 32: The rationale for binarizing Likert-scale ratings (collapsing into "help" vs. "no help") is noted. However, treating willingness-to-help as a continuous measure might better capture nuanced decision thresholds. Could the authors also model the raw ratings to assess robustness? A comparison of results using both approaches would strengthen methodological transparency.

We agree that comparing results from both approaches will enhance methodological transparency. We thus also fit WTH ratings (-5 to 5) using the same primary model from the manuscript and compared model fits and parameter estimations from both versions. These two modeling results are highly correlated with each other in the measures across participants: (1) the model fits (mean squared error in the continuous model, and negative log-likelihood in the binary model, $R = 0.657$), (2) fitted parameter weight of self-cost ($R = 0.704$), (3) fitted parameter helping bias ($R = 0.881$). In addition, we replicated the model sanity checks in Figure 5 c-f, and found the same qualitative results in the new model. Please find details in Supplementary Figure 5 of the current manuscript.

*(page 38) “**Model robustness check***

Although our primary focus is to model the help vs. not help decision as a binary variable, we additionally replicated the selected model (2.0) with a continuous rating version to check the robustness of the model fitting results. This continuous version (hereafter named 'rating model') was coded the same way as the binary version (hereafter named 'decision model'), except that (1) we applied a scaling factor to scale up the estimated utility to match the range of WTH rating rather than calculating a probability with the softmax function, and (2) we optimized through minimizing the mean squared error (MSE) between the predicted rating and actual ratings.

We compared the rating model with the decision model in multiple aspects. First, to check whether the rating model can obtain similar model fitting results, we correlated the model performance metric (MSE for the rating model, and log likelihood for the decision model) and estimated parameters (w_{cost} , b) between the two models. Second, we checked whether the rating model showed similar predictive properties by calculating (1) the out-of-sample predictions of WTH

rating (to replicate Figure 3f), (2) correlation between w cost and WTH ratings under high vs. low cost conditions (to replicate Figure 3c), (3) correlation between w cost and WTH ratings under high vs. low benefit conditions (to replicate Figure 3d), and (4) correlation between b and WTH (to replicate Figure 3e). All the above analyses confirmed that the rating model showed highly consistent results with the decision model (see details in Supplementary Figure 5), suggesting high robustness of the original modeling approach.”

Supplementary Figure 5. Comparison between the binary decision model and the continuous rating model. A continuous version of the model 2.0 (rating model) that fit to the WTH ratings instead of binary choices (decision model) showed highly consistent (a) model fits (LLH: log-likelihood, MSE: mean squared error), (b) fitted parameter of w_{cost} , and (c) fitted parameter of b compared to the binary version. It also replicated multiple properties of the model, including (d) good out of sample prediction accuracy on the WTH ratings (replicating Figure 3f), (e) positive correlation between the weight of self cost (w_{cost}) and differences in WTH between low and high cost scenarios (replicating Figure 3c), (f) negative correlation between the weight of self cost (w_{cost}) and differences in WTH between high and low benefit scenarios (replicating Figure 3d), and (g) positive correlation between helping bias (replicating Figure 3d) (b) and WTH ratings (replicating Figure 3e).

3. Page 31: The choice of varimax rotation assumes orthogonal (uncorrelated) factors. Given that motivations like "cost" and "benefit" may theoretically

interrelate, could the authors consider promax rotation (allowing correlated factors) and report whether results meaningfully differ? This would ensure the factor structure optimally reflects potential motivational interdependencies.

Thank you for bringing this to our attention. We agree with this concern, and indeed, we considered this during our initial analysis. We performed both varimax rotation and promax rotation, the latter allows correlated factor structures. We decided to report results from the varimax rotation because the promax rotation suggested that the two factors are uncorrelated ($r=0.01$). Comparison of the factor loading results between the two methods also showed highly consistent factor structures (Factor 1 correlation between 2 methods = 0.996, Factor 2 correlation between 2 methods = 0.984). We have added relevant information to the manuscript to increase the transparency of our methodological decisions.

(page 34, Methods) “As the parallel analysis output 2 factors, we conducted the EFA with a 2-factor structure using the varimax rotation (assuming an orthogonal factor structure). Note that the use of a promax rotation (that allows correlation between factors) generated highly consistent factor loadings (Correlation between Varimax and Promax loadings across items: Factor 1: $R = 0.996$, Factor 2: $R = 0.984$, Supplementary Figure 3e) and also suggested minimal correlation ($R=0.01$) between the 2 factors, thus in subsequent analyses, we proceed with the varimax rotation results.”

Supplementary Figure 3. Exploratory factor analysis on the original motivation dimensions. (e) Comparison of factor loadings between promax and varimax rotation results. Each dot represents a motivation dimension.

4. The analyses linking decision, motivation, and semantic spaces are technically sound. To enhance conceptual clarity, the manuscript might more

explicitly state how these associations provide novel insights beyond confirming expected relationships between each latent spaces.

Thank you for raising this concern. We recognize the need to improve the clarity of this section. We therefore revised the description about the representational similarity analyses, included an additional figure to facilitate understanding, and more explicitly highlight the implications of these analyses.

(p18, Results) "The above results indicate that different benefit and cost profiles in different semantic categories are associated with the differences of overall WTH across the scenarios. However, even when two scenarios are judged with the same average WTH, they may not be evaluated in the same way by each individual (see further explanation in Supplementary Figure 8). What accounts for this within-rater consistency? To answer this question, we partitioned the scenario decision similarities into two components by decomposing the decision space representational dissimilarity matrix (RDM) to a WTH RDM (i.e., an RDM calculated based on the mean WTH rating differences) and a residual RDM (that accounts for additional variances due to within-rater variability). We found that while the WTH RDM was significantly predicted by benefit ($\beta = 0.476, s.e. = 0.012, p < 0.001, 95\%CI = [0.452, 0.501]$) and cost ($\beta = 0.229, s.e. = 0.013, p < 0.001, 95\%CI = [0.205, 0.254]$) RDM (but not the semantic space RDM) the residual RDM was significantly predicted by the semantic space RDM alone ($\beta = 0.070, s.e. = 0.015, p < 0.001, 95\%CI = [0.041, 0.098]$). These results suggested that although the average group consensus of WTH level is associated with the costs and benefits, scenarios from the same semantic categories are evaluated more consistently within an individual. In other words, individuals tend to maintain consistent criteria when judging scenarios of the same semantic type."

(Discussion) "Leveraging the cross-scenario similarities derived from the three spaces, we further dissociated the unique contributions of the motivation space and semantic space in driving helping behaviors: across individuals, consensus on the costs and benefits is related to consensus on the willingness to help, whereas within individuals, semantically similar situations are evaluated more consistently."

Supplementary Figure 8: Decomposition of the decision similarity space. The decision similarity space quantifies two levels of similarity: similarity in consensus ratings of WTH (WTH RDM), and within-individual consistencies of ratings among scenarios (residual RDM). In this illustration, even though scenario x, y, and z all have an average rating of 2, scenario x is more similar to z than y because x and z are rated in the same way across three raters.